



# Variability of ice supersaturated regions at flight altitudes: evaluation of ERA5 reanalysis using IAGOS in situ measurements

Katarina Grubbe Hildebrandt[1], Federica Castino[1], Vincent Meijer[1], and Feijia Yin[1]

[1]Delft University of Technology, Faculty of Aerospace Engineering, Delft, The Netherlands

**Correspondence:** Katarina Grubbe Hildebrandt (k.g.hildebrandt@tudelft.nl)

**Abstract.** Contrail cirrus is one of the largest contributors to aviation's radiative forcing, which arises from long-lived persistent contrails. Avoiding persistent contrail formation has been suggested as a measure to reduce the climate impact of aviation, requiring accurate forecasts of ice supersaturated conditions, i.e. where the relative humidity over ice (RHi) exceeds 100%. Numerical weather prediction (NWP) models often underestimate or do not account for ice supersaturation. This study evaluates ice supersaturated regions (ISSRs) in the ECMWF ERA5 reanalysis using In-service Aircraft for a Global Observing System (IAGOS) measurements over tropical and extratropical regions in the upper troposphere and lower stratosphere for the period 2011-2022. It considers cloudy and clear-sky conditions and how North Atlantic weather patterns affect the ERA5 ability to predict ISSRs. ERA5 generally underestimates ISSR occurrence due to a dry bias in RHi; the equitable threat score (ETS) is 0.2-0.4, indicating a weak to mediocre relationship with IAGOS. Clear-sky conditions result in an ETS of 0.05-0.18 and generally below 0.1 in cloudy conditions, indicating an almost random relationship. The latter is the result of the saturation adjustment used by the NWP model underlying the reanalysis. North Atlantic winter weather patterns appear to affect the ability of ERA5 to predict ISSRs, particularly along eastbound routes. This may result from varying ISSR distributions relative to the jet stream. North Atlantic summer weather patterns show little impact due to weaker teleconnection patterns. Overall, the underestimation of ISSRs in ERA5 is most critical in the upper troposphere, where their occurrence is highest.

## 1 Introduction

The aviation industry is an important contributor to anthropogenic climate change. In 2018, aviation accounted for 2.5% of the world's $CO_2$ emissions (Lee et al., 2021; Teoh et al., 2022; Wolf et al., 2023). Aviation has also been estimated to contribute to 3.5% to 5% of global anthropogenic radiative forcing (Lee et al., 2021). The anthropogenic radiative forcing is due to $CO_2$ and non-$CO_2$ emissions, which include $NO_x$ emissions, $H_2O$, soot, contrails, and contrail cirrus (Lee et al., 2009). The best estimate of contrail cirrus effective radiative forcing is almost twice as large as that of $CO_2$, but is also subject to much larger uncertainties (Lee et al., 2021). This is due to a large number of sources of uncertainty in the evaluation of contrail cirrus (Wilhelm et al., 2021).

One solution to lowering the climate impact of aviation is to minimise the radiative forcing due to contrail cirrus. Contrail cirrus is the result of the dispersion of persistent contrails. Contrails form when the Schmidt-Appleman criterion (SAC) is met (Schumann, 1996; Gierens et al., 2020a), and persist when the ambient air is supersaturated with respect to ice, i.e. relative



humidity over ice (RHi) is greater than 100% (Gierens et al., 2020a; Wolf et al., 2025). A region where the latter occurs is called an ice supersaturated region (ISSR) (Reutter et al., 2020). Reducing persistent contrails can be realised by avoiding flying through ISSRs (Mannstein et al., 2005; Filippone, 2015; Yin et al., 2018; Avila et al., 2019; Teoh et al., 2020; Martin Frias et al., 2024; Sausen et al., 2024; Sonabend-W et al., 2024). This requires accurate predictions and evaluations of ice supersaturation

(ISS). However, ISS is often not accounted for or underestimated in numerical weather prediction (NWP) models (Rädel and Shine, 2010; Gierens et al., 2020a; Wilhelm et al., 2022; Agarwal et al., 2022; Wolf et al., 2023).

The lack of ISS in NWP models has been shown to arise due to several reasons. One reason is the large temporal and spatial variability of the humidity field, with sharp gradients (Wilhelm et al., 2022; Sperber and Gierens, 2023; Wolf et al., 2025). It is also due to a lack of reliable relative humidity measurements at aircraft cruise altitudes (Sperber and Gierens, 2023) and due

to the coarse resolution of weather models (Gierens et al., 2012). This leads to biases in the prediction of relative humidity in weather models. Reutter et al. (2020) showed that ERA-Interim, the predecessor of ERA5, underestimated RHi when greater than 100%, leading to an underestimation in the occurrence of ISS when compared to MOZAIC (Measurement of OZONE and Water Vapour on Airbus in-service Aircraft). It has also been shown that ERA5 has a dry bias when RHi is above 100%, when compared to MOZAIC/IAGOS (In-Service Aircraft for Global Observing system) (Gierens et al., 2020a; Schumann et al.,

2021; Teoh et al., 2022; Wolf et al., 2025). Contrarily, Dyroff et al. (2014), Shepherd et al. (2018) and Bland et al. (2021) have shown a moist bias in the lower stratosphere of the ECMWF Integrated Forecasting System (IFS). Meanwhile, there are often good agreements in temperature between model predictions and measurements (Dyroff et al., 2014; Reutter et al., 2020; Wolf et al., 2025).

Another reason for the underestimation of RHi in NWP models is related to the representation of the physics of the cloud

nucleation process (Dyroff et al., 2014). Previously, supersaturation with respect to ice was not allowed in models such as the ECMWF IFS (Dyroff et al., 2014). However, the ECMWF IFS now uses the saturation adjustment in the ice cloud microphysical scheme to allow for ISS (Tompkins et al., 2007; Wolf et al., 2025). The saturation adjustment can be explained as when cloud formation occurs in an ice supersaturated grid box, RHi is lowered to 100% within the cloudy part of the grid-box in the next time step (Tompkins et al., 2007; Straka, 2009; Sperber and Gierens, 2023). In clear-sky conditions, models such as the

IFS, can also represent ISS (Tompkins et al., 2007; Reutter et al., 2020). However, ERA5, generated with the ECMWF IFS Cycle 41r2, shows a lack of ISS under (almost) clear-sky conditions in the mid-latitudes, as reported by Wolf et al. (2025). Wang et al. (2025) also showed that ERA5 was limited in predicting ISSRs under clear-sky conditions. Hence, issues in the representation of ISS occurs in both cloudy and clear-sky conditions in ERA5.

Many of the studies analysing the accuracy of NWP models in prediction of ISS compared to observations, i.e. MOZAIC/I-

AGOS, are often limited in terms of regional and seasonal variations. The main area of interest are the mid-latitude regions as this is one of the most sampled regions by MOZAIC / IAGOS aircrafts (Reutter et al., 2020; Sanogo et al., 2024; Wolf et al., 2025). Only Gierens et al. (2020a) compared IAGOS and ERA5 below a latitude of 30°, considering four months in 2014. The entire MOZAIC/IAGOS framework now spans more than 20 years (Reutter et al., 2020). Furthermore, there are limited comparisons of RHi and ISS between the subregions of the globe. In some instances, this is due to a smaller focused area,

i.e. the North Atlantic corridor. Reutter et al. (2020) selected a larger region, considering North America, the North Atlantic





corridor, and Europe, but the analysis on regional variations were limited. In case of the study by Gierens et al. (2020a), variations due to different climates (tropics versus extratropics) is not included. Thus, there is a need to further quantify regional, including climate, dependence of the differences in ISS between the weather model predictions and observations, in particular, the IAGOS measurements.

Additionally, previous research shows that the occurrence of ISSRs has a seasonal dependence in the northern mid-latitudes (Petzold et al., 2020) and over the Paris area (Wolf et al., 2023). Gierens et al. (2012) also showed a seasonal variation in ISSR occurrence, with less ISSRs from April to September over the Lindenberg meteorological observatory. Petzold et al. (2020) showed varying ISSR and RHi seasonality between Europe, the North Atlantic Corridor and eastern North America. However, the seasonal differences in ISSR occurrence between observations and NWP models are rarely considered. Gierens

et al. (2020a) analysed seasonal differences in RHi and ISS between IAGOS observations and the ERA5 reanalysis considering four months of one year of data, with each month representing a season; some seasonal dependency on the differences between IAGOS and ERA5 was identified. Since the study only considered a four month dataset, it has a high potential to be expanded with extensive input data. Also, no analysis of seasonal differences as a function of geographic region appears to have been performed. Hence, it is important to further quantify these differences between observations and weather models.

A region of particular interest is the North Atlantic corridor, one of the busiest air traffic corridors (Teoh et al., 2022). Irvine et al. (2012) found that the probability of forming contrails as a function of altitude depends on the weather patterns when using ERA-Interim. This raises the question of whether the type of weather pattern can affect the observed biases in ERA5 as well.

    Therefore, the aim of this study is to further investigate differences in ISS between the ECMWF ERA5 reanalysis and

IAGOS in situ measurements, with an expanded geographical area. Differences will be analysed from a regional and seasonal perspective, to understand if certain conditions might lead to a drier or more moist bias in ERA5. The impact of clouds on differences in ISS between IAGOS and ERA5 will also be analysed on a larger geographical scale. Lastly, it will be investigated if different winter and summer weather patterns in the North Atlantic corridor impact the ability of ERA5 to predict ISSRs.

    Section 2 describes the datasets and the methodology used in this study. Section 3 documents differences in temperature and

RHi and the impact of the prediction of ISSRs. This section also documents the influence of different weather conditions on the ability of ERA5 to predict ISS. Lastly, conclusions are drawn in Sect. 4.

## 2   Data and methodology

This section explains the data selection from the IAGOS in situ measurements and the method to evaluate the ERA5 reanalysis.

### 2.1   IAGOS

IAGOS (Petzold et al., 2015) provides atmospheric composition measurements from commercial aircraft and was founded in 2011 (IAGOS, last access: 2025-04-15a). It is the successor of the MOZAIC programme and the Civil Aircraft for the Regular Investigation of the Atmosphere Based on an Instrument Container (CARIBIC), which began measurements in 1994.



Meanwhile, MOZAIC is no longer a project, CARIBIC continues within IAGOS and serves as a reference standard for the rest of the IAGOS fleet (Petzold et al., 2015).

The IAGOS-CORE component is installed on long-range aircraft from internationally operated airlines (Petzold et al., 2015), where 10 aircraft are currently active (IAGOS, last access: 2025-04-15d). IAGOS-CORE contains several autonomous instruments for daily measurements of reactive, i.e. $O_3$, and greenhouse gases, such as $CO_2$ and water vapour, as well as aerosol and cloud particles (Petzold et al., 2015). The main package installed in the entire IAGOS fleet is Package 1, which includes a humidity sensor (ICH) and a backscatter cloud probe (BCP) that measures cloud particles (IAGOS, last access: 2025-04-15c).

The ICH consists of a modified Vaisala HUMICAP® of type H sensor (capacitive relative humidity sensor) (Patrick Neis and Herman G. J. Smit and Susanne Rohs and Ulrich Bundke and Martina Krämer and Nicole Spelten and Volker Ebert and Bernhard Buchholz and Karin Thomas and Andreas Petzold) and a platinum resistance sensor for temperature measurements at the surface of the humidity sensing element (IAGOS, last access: 2025-04-15b). It has the capability to provide temperature and relative humidity (with respect to liquid) measurements. The relative humidity has a precision of ± 1%, an accuracy of ±

6% and a time resolution of 1 s at 300 K to 120 s at 200 K (IAGOS, last access: 2025-04-15b). Temperature measurements have a precision of ± 0.2 K, an accuracy of ± 0.5 K and a time resolution of 4 s (IAGOS, last access: 2025-04-15b).

Given the start date of the IAGOS program and that the entire IAGOS fleet was equipped with the modified Vaisala HU-MICAP® of type H sensor in 2011 (Patrick Neis and Herman G. J. Smit and Susanne Rohs and Ulrich Bundke and Martina Krämer and Nicole Spelten and Volker Ebert and Bernhard Buchholz and Karin Thomas and Andreas Petzold), data measured

between July 2011 and December 2022 have been collected for the current study. The end date is set to December 2022 due to a lack of published flight measurements with all observed variables after this date (at the time of data collection). The altitude is restricted between 8000 m and 13000 m as these are the flight levels that are the most visited by IAGOS aircraft. These are also the levels where we can expect ISSRs to occur in the midlatitudes (Lamquin et al., 2012; Sanogo et al., 2024). In the tropics, ISSRs are rare below 10000 m, and have been show to occur at altitudes of more than 15000 m (Lamquin et al., 2012).

Commercial aircraft do not cruise at such high altitudes, as also seen from the IAGOS dataset. Hence, we maintain the upper bound of 13000 m.

Based on the global distribution of the IAGOS measurements during the considered time frame, as seen in Fig. 1, only measurements located between 160°W and 150°E, and between 30°S and 75°N will be considered. This is further divided into eight regions. The latitudes of the regions are based on the definition of tropical (30°S ≤ latitude ≤ 30°N) and extratropical

regions (latitude < 30°S and latitude > 30°N) (Spichtinger and Leschner, 2016). The eight regions are shown in Fig. 1 and have the areal boundaries given in Table 1.

Only points with a relative humidity with respect to liquid (RHL) between 0% and 100% are selected from IAGOS measurements. A RHL greater than 100% implies flying through a liquid cloud, and contrails cannot form at temperatures where these clouds are present (Wilhelm, 2022).

IAGOS uses validity flags to indicate the quality of the measurements. The following validity flags are used: 'good' (0), 'limited' (2), 'erroneous'(3), 'not validated' (4) and 'missing value' (7) (IAGOS, last access: 2025-06-10). Only 'good' and 'limited 'measurements are considered for temperature, relative humidity with respect to ice (RHi) and RHL. However, we





use the RHL validity to select RHI values since the quality flag of RHi is not well derived, but is known to be similar to RHL (Sanogo et al., 2024). A summary of the criteria for the IAGOS measurements is provided in Table 2.

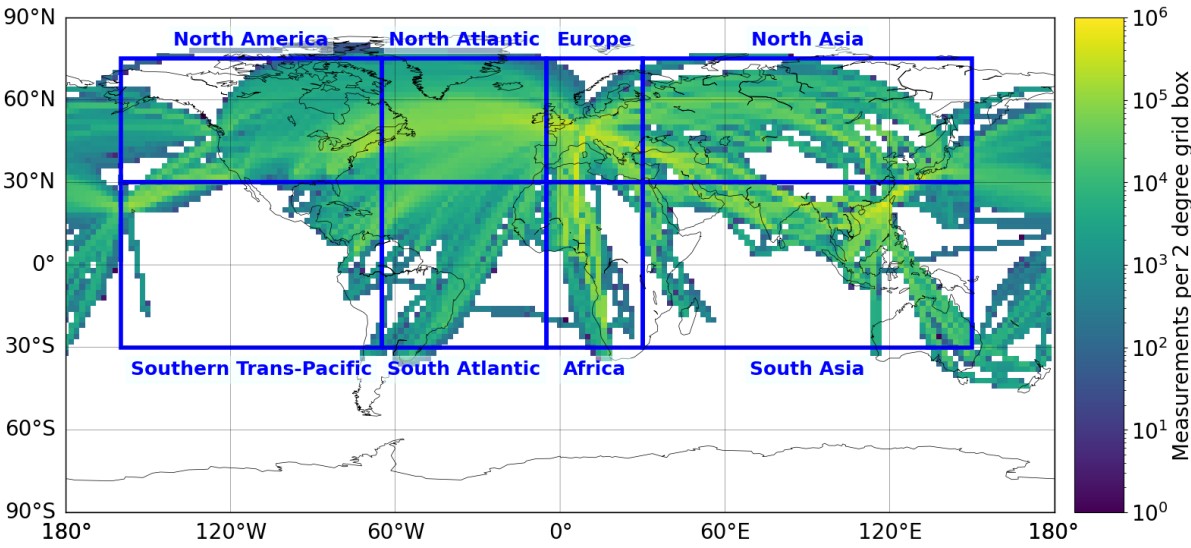

**Figure 1.** IAGOS measurement density map per 2°grid box with regional division.

**Table 1.** Areal boundaries of defined geographical regions.

| Region | Longitude | Latitude |
|---|---|---|
| North America | 160°W to 65°W | 30°N to 75°N |
| North Atlantic | 65°W to 5°W | 30°N to 75°N |
| Europe | 5°W to 30°E | 30°N to 75°N |
| North Asia | 30°E to 150°E | 30°N to 75°N |
| Southern Trans-Pacific | 160°W to 65°W | 30°S to 30°N |
| South Atlantic | 65°W to 5°W | 30°S to 30°N |
| Africa | 5°W to 30°E | 30°S to 30°N |
| South Asia | 30°E to 150°E | 30°S to 30°N |

Lastly, IAGOS records the measurements every four seconds. To avoid autocorrelation, it is chosen to sample approximately every minute, which corresponds to 2.5% of all IAGOS measurements between 01/07/2011 and 31/12/2022. To avoid systematic bias, the data points are randomly sampled using a uniform random number generator ranging from 1 to the maximum number of measurement points from IAGOS.



**Table 2.** Criteria for selection of IAGOS measurements.

| Criterion | |
|---|---|
| **Measurement dates** | Between 01/07/2011 and 31/12/2022. |
| **Altitude** | Between 8000 m and 13000 m. |
| **RHL** | Between 0% and 100%. |
| **Validity flags for RHi, RHL and temperature** | Equal to 0 or 2, corresponding to 'good' or 'limited'. |
| **Geographic coverage of measurements** | Located between 160°W and 150°E, and between 30°S and 75°N |

## 2.2 ERA5 reanalysis

The ERA5 reanalysis is the fifth generation reanalysis from ECMWF (Hersbach et al., 2020). To allow for a higher vertical resolution, the Analysis-Ready, Cloud Optimized (ARCO) ERA5 (Carver and Merose, 2023) has been used, which provides several atmospheric variables on model levels. ARCO ERA5 was retrieved using the python library pycontrails (Shapiro et al., 2025), which also allows for the re-gridding of model levels to pressure levels. It was chosen to re-grid with a vertical resolution of 10 hPa. This is within the range of the model level resolution at typical cruise altitudes, which is between 8 hPa and 15 hPa

when considering a surface pressure of 1013.250 hPa (ECMWF, last access: 2025-05-07).

The ERA5 reanalysis is (quadrilinearly) interpolated in time, pressure level, longitude and latitude onto the IAGOS flight tracks using the Delftblue supercomputer (Delft High Performance Computing Centre, 2024). The ARCO ERA5 dataset does not provide RHi and RHL, hence specific humidity, temperature and saturation water vapour pressure over liquid water and ice are used to calculate these two variables. However, specific humidity exhibits a nonlinear lapse rate (pycontrails, 2025). Thus,

to avoid biases due to linear interpolation, RHL and RHi should be calculated before interpolation, where we then interpolate in the RHi and RHL space (pycontrails, 2025).

## 2.3 Vertical distribution of variables with respect to tropopause

The vertical distribution of the temperature, relative humidity over ice, and ISSR fraction will be reported relative to the tropopause height. Both the dynamic and thermal tropopause have been considered, and were determined using ERA5 tropo-

spheric data reported by Hoffmann and Spang (2022). Given the stronger gradients in temperature and RHi associated with the thermal tropopause (see Appendix A), the results are presented relative to this definition. This will, for example, result in a drier lower stratosphere compared to if the dynamic tropopause definition was used.

The flight measurements are distributed into layers with a thickness of 30 hPa, centered at the tropopause. The selection of layers for further analysis depends on the number of data points per level, which varies per geographical region, as seen in

Fig. 2. This figure shows that in the extratropic regions, the majority of measurements are distributed around the tropopause, allowing for upper tropospheric and lower stratospheric analysis. In the tropics, most of the samples are located in the upper troposphere since the tropopause is located at higher altitudes in this region.





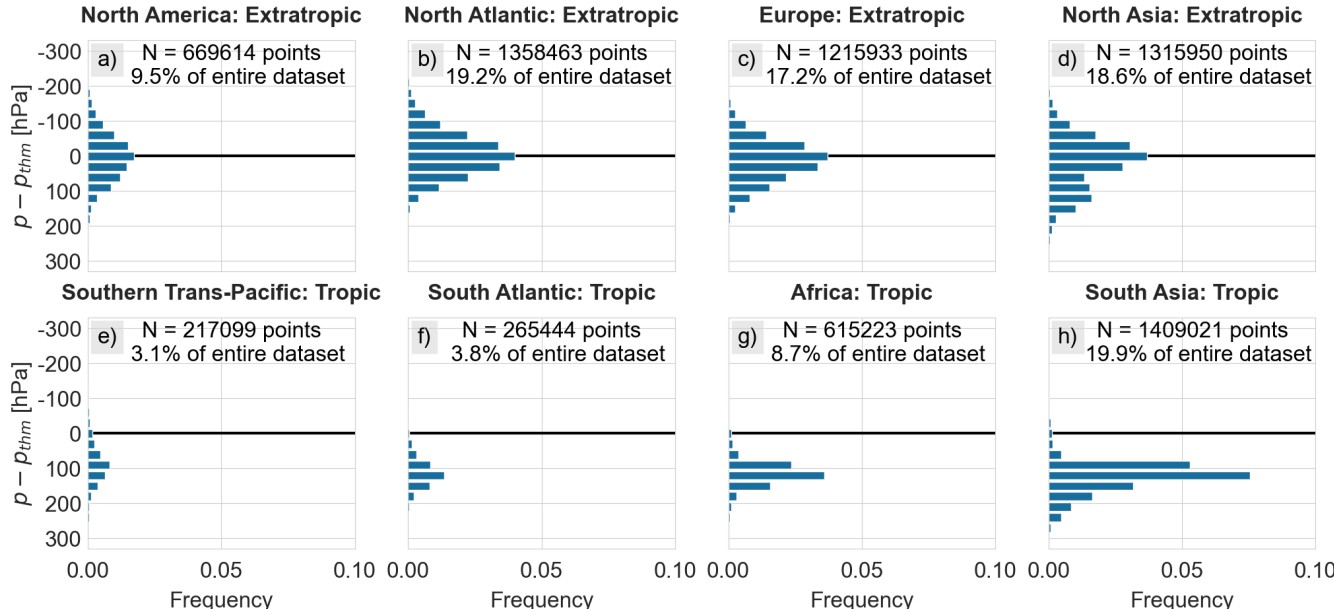

**Figure 2. (a-h)** Vertical distribution of number of samples with respect to thermal tropopause, $p_{thm}$, in different geographical regions.

## 2.4 Differentiation of cloudy and clear-sky conditions

To investigate the impact of cloudy and clear-sky conditions on the ability of ERA5 to predict ISS, we need to distinguish
between these two conditions. For IAGOS, the number of ice particles, $N_i$, can be used to differentiate between measurements
within cirrus clouds and within cloud-free conditions. We use the same thresholds defined by Petzold et al. (2017), which can
be summarised as follows:

- Cloudy: $N_i \geq 0.015 \, \mathrm{cm}^{-3}$

- Clear-sky: $N_i \leq 0.001 \, \mathrm{cm}^{-3}$

- Indeterminate: $0.001 < N_i < 0.015 \, \mathrm{cm}^{-3}$. This refers to measurements that cannot be identified as clouds and cannot be
considered cloud-free. These measurements are most likely located within very thin cirrus (Petzold et al., 2017).

The fraction of cloudy, clear-sky and indeterminate conditions, based on the definition of the number of ice particles, per
layer and geographical region is visualised in Fig. 3. The majority of samples are categorised as no clouds or no measurements.
There are few indeterminate and cloudy conditions. The low number of cloud observations is in line with reports from Sanogo
et al. (2024).



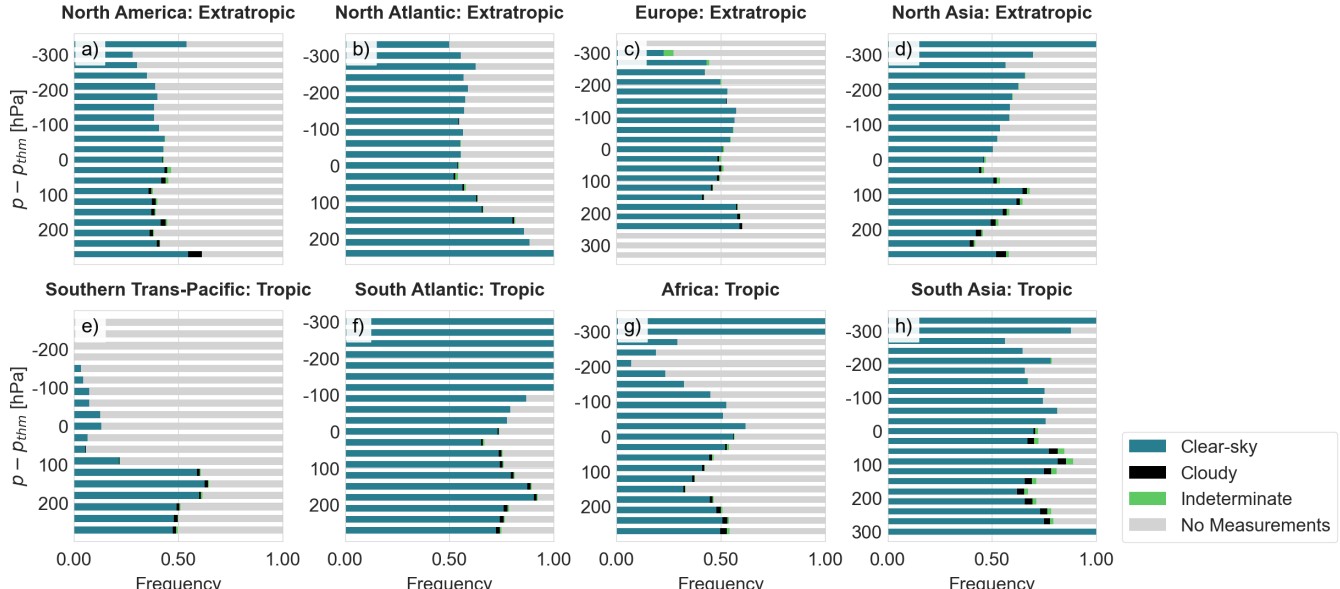

**Figure 3. (a-h)** Fraction of cloudy, clear-sky and indeterminate conditions per layer and geographic region, based on the number of ice particles, $N_i$, from IAGOS measurements.

For the ERA5 reanalysis, cloudy and clear-sky conditions can be differentiated using the cloud cover fraction, $CC$, which takes a value between 0 and 1. $CC$ describes the proportion of a grid box covered by liquid or ice clouds (Hersbach et al., 2023a). For the analysis of RHi and ISSRs, we only consider data points where the temperature is below 235.15 K, which is below the threshold at which liquid clouds can occur (Gierens et al., 2020b), hence we focus on ice clouds. To distinguish

between cloudy, clear-sky and indeterminate conditions, we use the same thresholds as Wolf et al. (2025):

  – Cloudy: $0.8 \leq CC \leq 1$

  – Clear-sky: $CC < 0.2$

  – Indeterminate: $0.2 \leq CC < 0.8$

The application of cloud cover to determine cloudy, clear-sky and indeterminate conditions using ERA5 results in the divi-

sion seen in Fig. 4. We applied the 'no measurements' label to ERA5 for the same points as in IAGOS.

Comparing Fig. 3 and Fig. 4, it is noticeable that there are discrepancies in the labelling of cloudy, clear-sky and indeterminate conditions between IAGOS and ERA5. We see that ERA5 shows a higher frequency of indeterminate and cloudy conditions compared to IAGOS.





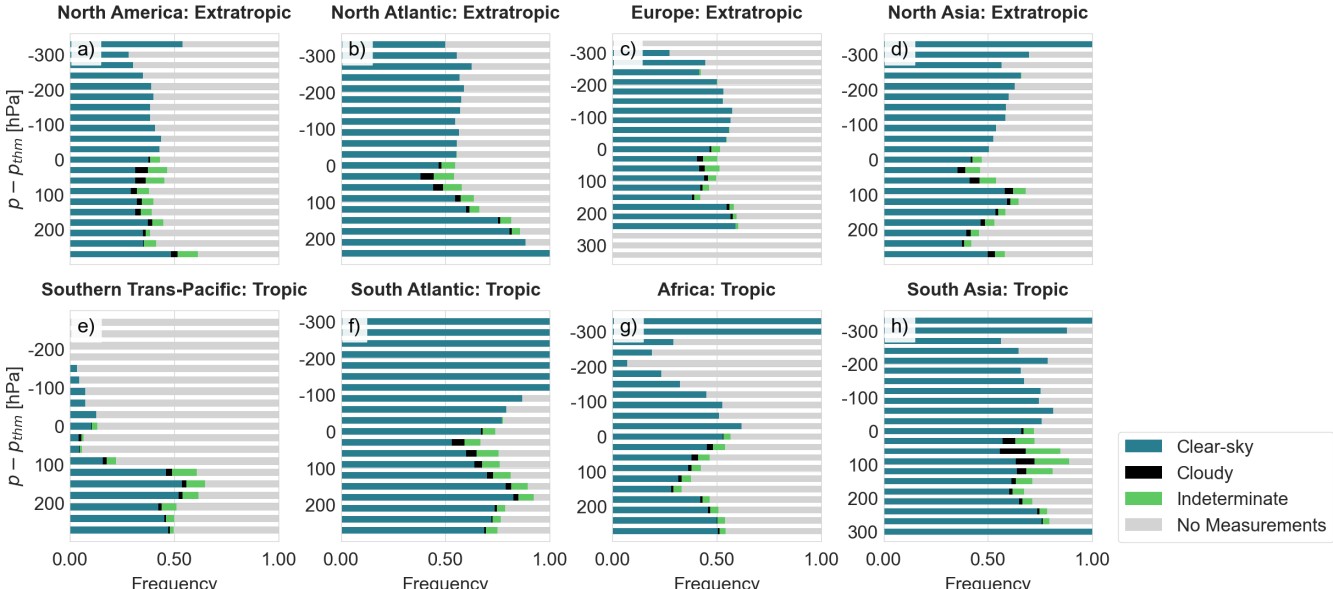

**Figure 4. (a-h)** Fraction of cloudy, clear-sky and indeterminate conditions per layer and region, based on the definitions using the cloud cover from ERA5.

The mean annual global cirrus cloud occurrence using IAGOS measurements can be seen in Fig. 5. The location of large
amounts of cirrus corresponds well with cirrus hotspots identified by Petzold et al. (2017), albeit the percentages are lower. This may be the result of using a larger timeframe compared to Petzold et al. (2017) given that we use the same discretisation for the figure. Similar percentages are achieved when considering the overall relative cirrus occurrence.

## 2.5    North Atlantic weather pattern classification

The North Atlantic weather patterns are determined using the classification presented by Irvine et al. (2013), which is based
on the similarity of the daily mean geopotential height anomaly to typical patterns, i.e. the North Atlantic Oscillation (NAO) and East Atlantic (EA) patterns. The daily mean geopotential height is obtained from the Copernicus Climate Data Store (CDS) (Hersbach et al., 2023b) at a pressure level of 250 hPa. Subsequently, the anomalies with respect to the entire period 2011-2022 are calculated. Days are then assigned to five winter weather patterns (W1-W5) and three summer weather patterns (S1-S3), according to a set of criteria (Irvine et al., 2013). The characteristics of these weather patterns can be found in Table 3.
The frequency of these patterns is presented in Fig. 6. The average number of days per season within each weather pattern class aligns well with the literature (Irvine et al., 2013). The analysis in Sect. 3.6 considers the extended winter season, from December to March.





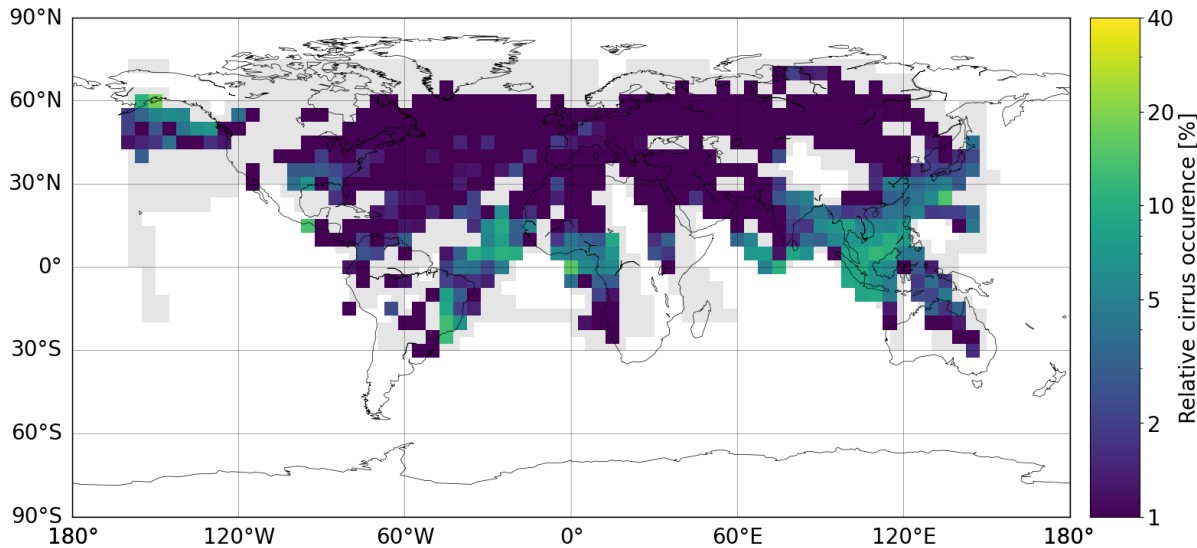

**Figure 5.** Mean annual cirrus cloud occurrence using IAGOS number of ice particles for all considered vertical levels and measurements between July 2011 and December 2022. Light grey area is the entire coverage of the IAGOS flights considered in this study, for which there are no measurements for number of ice particles.

**Table 3.** North Atlantic weather pattern characteristics for winter (W1-W5) and summer (S1-S3) (Irvine et al., 2013).

| Type | Pattern | Jet stream position | Jet stream strength |
|------|---------|---------------------|---------------------|
| W1 | EA+ | Zonal | Strong |
| W2 | NAO+ | Tilted | Strong |
| W3 | EA- | Tilted | Weak |
| W4 | NAO- | Confined | Strong |
| W5 | Mixed | Confined | Weak |
| S1 | EA+ | Zonal | Strong |
| S2 | Mixed | Weakly tilted | Weak |
| S3 | EA- | Strongly tilted | Weak |





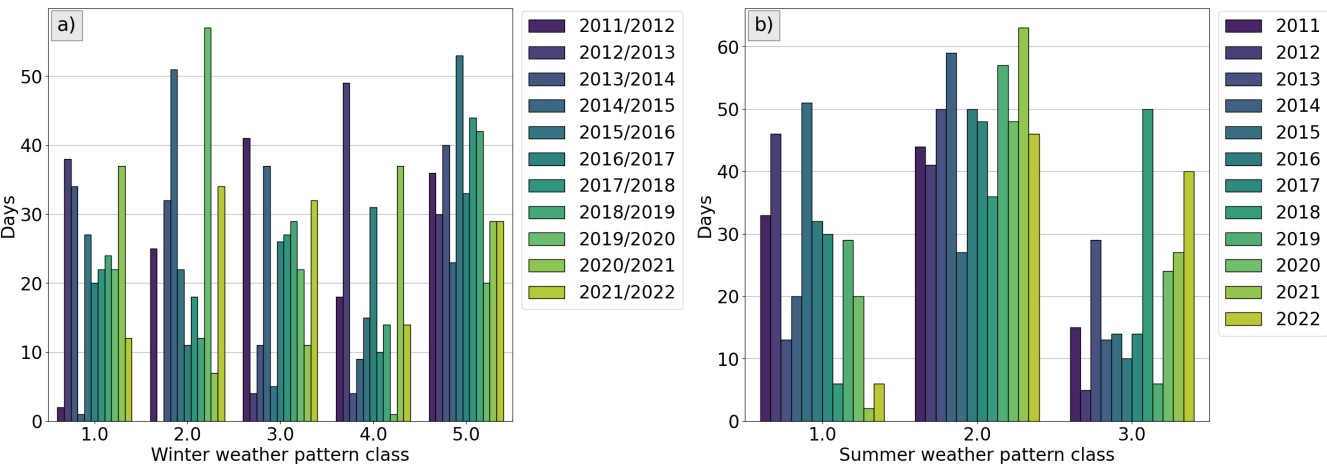

**Figure 6.** Frequency of **a)** winter weather patterns and **b)** summer weather patterns per year (2011-2022) using ERA5.

## 3 Results

The aim of this section is to compare temperature, RHi and ISSR occurrence between IAGOS and ERA5.

### 3.1 Distribution of temperature from IAGOS and ERA5

As a first step, we compare the IAGOS measured temperature and the ERA5 simulated temperature. This is important to consider, as relative humidity partially depends on temperature (Reutter et al., 2020). Figure 7 shows the vertical mean temperature distribution of IAGOS and ERA5 per season, for the eight geographical regions defined. The choice for the lower limit on the number of samples is explained in Sect. S1 in the Supplement. Overall, there is good agreement in temperature, though ERA5 tends to have a cold bias in the extratropics, both in the upper troposphere and lower stratosphere. This is in line with results presented by Wolf et al. (2025) for mid-latitudes at different pressure levels. The cold bias observed in the upper troposphere may be due to sensor error as the mean difference, around 0.5 K, is within the accuracy of the IAGOS temperature sensor, as can be observed in Fig. S5 in the Supplement. Here, it can also be seen that the lower stratosphere shows a slightly larger cold bias compared to the upper troposphere, with mean differences ranging from 0.75 K to 1 K and can thus not be explained by sensor accuracy alone. Shepherd et al. (2018) also reported a cold bias of up to 0.5 K in comparison to radiosondes in the lower stratosphere of ERA5 due to an underlying cold bias in the IFS.

Figure 7 also shows the extratropical seasonal cycle in temperature, with higher temperatures found in JJA and lower temperatures in DJF. However, temperature differences between IAGOS and ERA5 show no seasonal cycle, with similar mean differences observed between all four seasons, as seen in Fig. S5 in the Supplement. In North Asia, ERA5 shows a warm bias for season DJF until approximately 60 hPa below the tropopause, after which the mean difference approaches that of the other seasons. In the lower stratosphere, season DJF again starts to deviate from the other seasons, with the mean difference de-




creasing until ERA5 shows a warm bias again. However, given the mean differences are within the sensor accuracy of IAGOS, sensor error cannot be ruled out.

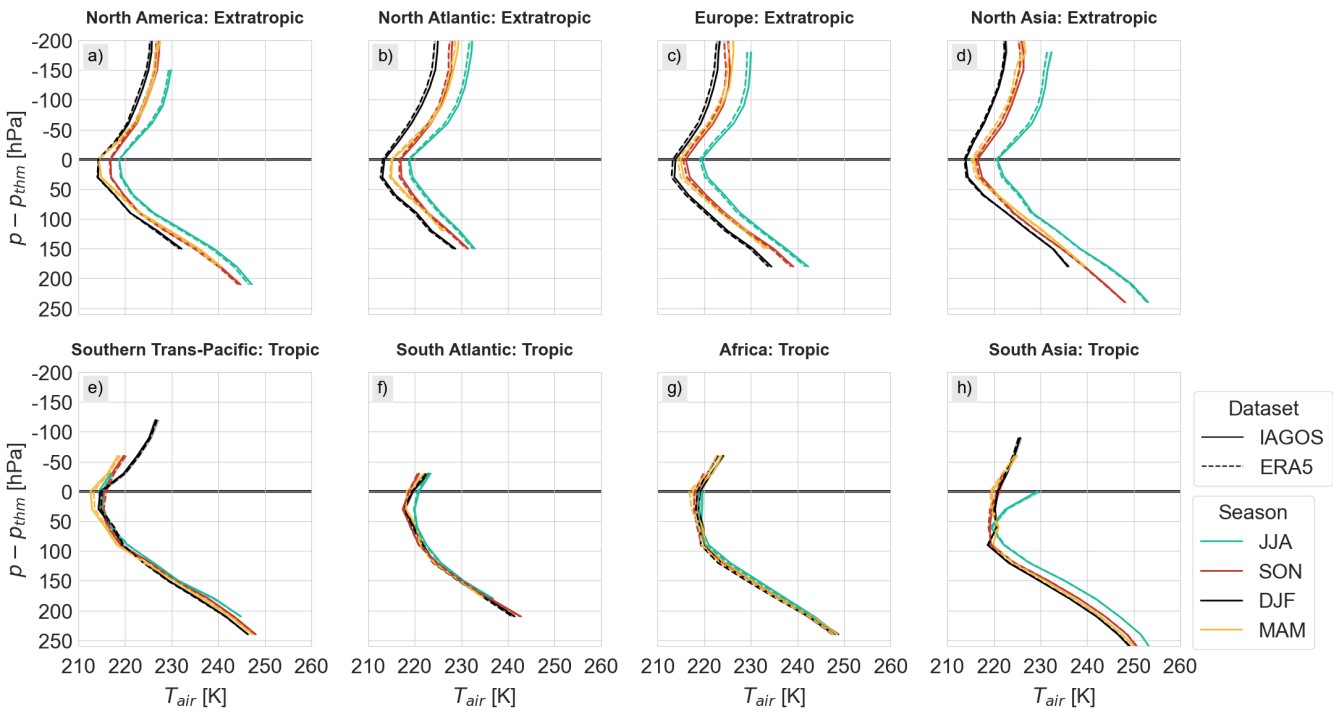

**Figure 7. (a-h)** Vertical distribution of IAGOS and ERA5 mean temperature per season and per region with shading showing the 95% confidence interval, using levels based on distance to thermal tropopause, only considering levels, seasons and regions with 500+ samples.

Meanwhile, the tropical regions show a better approximation of the temperature in ERA5 below the tropopause. The mean
temperature differences range between -0.25 K (ERA5 warmer than IAGOS) and 0.25 K, on average, in the upper troposphere. This is within the IAGOS temperature sensor accuracy range. In Africa, we see mean temperature differences between 0.5 K and 1 K. Furthermore, the tropics show an increased deviation between IAGOS and ERA5 close to and above the tropopause. In South Asia and Africa, the deviation causes a cold bias in ERA5, which can be attributed to the stratospheric cold bias in ERA5 (Shepherd et al., 2018). However, in the Southern Trans-Pacific, the increased deviation results in a warm bias, which
may be partially explained by sensor error. Generally, the minimum temperature is found at the thermal tropopause (Hoffmann and Spang, 2022), but in South Asia for season JJA, the minimum temperature is found approximately 60 hPa below the thermal tropopause. We conclude that this is not due to the following reasons as their consideration still results in the minimum temperature being located 30 hPa below the tropopause: the calculation of the tropopause or the temperature field in ERA5, outliers for specific years, and low sampling at the tropopause and in the lower stratosphere between 20°S and 20°N. We also
find that the minimum temperature occurs at the same distance from the dynamic tropopause. Hence, we hypothesise it may




be due to the type of weather encountered in season JJA in South Asia as deep convection can lead to perturbation of the temperature profiles with respect to altitude (Muhsin et al., 2018).

## 3.2 Distribution of relative humidity over ice from IAGOS and ERA5

Ice supersaturation is governed by the RHi. We only consider RHi for cases where the temperature is below the threshold of
homogeneous freezing, assumed equal to 235.15 K. This is the lowest temperature at which supercooling of cloud droplets can occur (Gierens et al., 2020b; Sperber and Gierens, 2023). Less than 5% of our IAGOS measurements show ISS above this temperature threshold.

Figure 8 shows a two-dimensional histogram, illustrating the overall ability of ERA5 to predict RHi. It shows a high frequency of points along the perfect agreement line. However, the highest frequency along this line occurs at low values of RHi,
which shows that ERA5 is more accurate at low values, as has also been observed by Wolf et al. (2025). There is also a high frequency of points along the RHi (ERA5) = 1 line due to the saturation adjustment in ERA5, in which the RHi is adjusted back to one when supersaturation occurs in a cloudy grid box, causing a concentration around this value. This is similar to the results obtained by Gierens et al. (2020a), Wolf et al. (2025), and Wang et al. (2025).

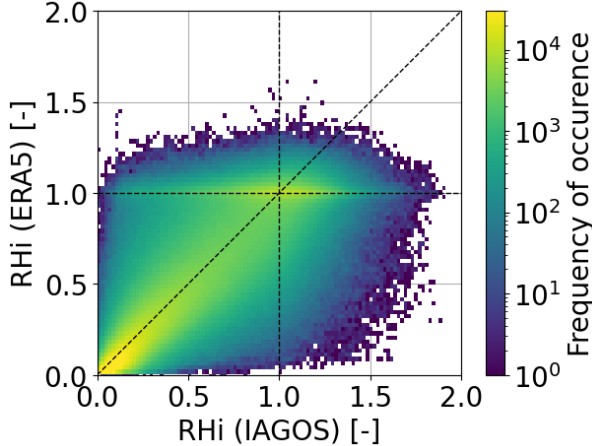

**Figure 8.** Two-dimensional histogram of ERA5 relative humidity over ice as a function of IAGOS relative humidity of ice.

The mean vertical RHi of IAGOS and ERA5 per season and geographic region is presented in Fig. 9. Generally, ERA5 shows
a dry bias, both in the upper troposphere and in the lower stratosphere. This is in line with other studies, such as Reutter et al. (2020) and Wolf et al. (2025). Some studies have reported a general moist bias in the lower stratosphere in the ECMWF-IFS (Dyroff et al., 2014; Shepherd et al., 2018; Bland et al., 2021), but we observe that the mean IAGOS RHi is higher than the mean ERA5 RHi in the lower stratosphere. However, we cannot ascertain that this is due to biases in ERA5. As explained by Wolf et al. (2025), low values of RHi measured by IAGOS in the lower stratosphere are subject to a moist bias. This results
from the limitation of the ICH sensor; it does not provide good quality results in dry conditions due to the loss of sensitivity as a result of the adiabatic compression effect (Konjari et al., 2025).





In the lower stratosphere, the range of RHi differences between IAGOS and ERA5 is smaller compared to the upper troposphere, as displayed in Fig. S6 in the Supplement. This is due to the dryness of the lower stratosphere, resulting in lower values of RHi as seen by the rapid decrease of RHi from approximately 30 hPa below the tropopause to 30 hPa above the tropopause,

which then changes to a near-vertical asymptote. This is more apparent in the extratropics, as the tropopause in the tropics is located at a higher altitude, which causes fewer samples in the lower stratosphere. Petzold et al. (2020) found the same vertical distribution of RHi with respect to tropopause layers in extratropic regions. The lower values of RHi results in ERA5 finding values of RHi closer to what has been observed with IAGOS. This was also shown in Fig. 8 with the large concentration of points along the perfect agreement line at low values of RHi. However, we acknowledge the possibility of a moist bias in

IAGOS, which means that ERA5 may show better ability of predicting RHi in the lower stratosphere than the results shown.

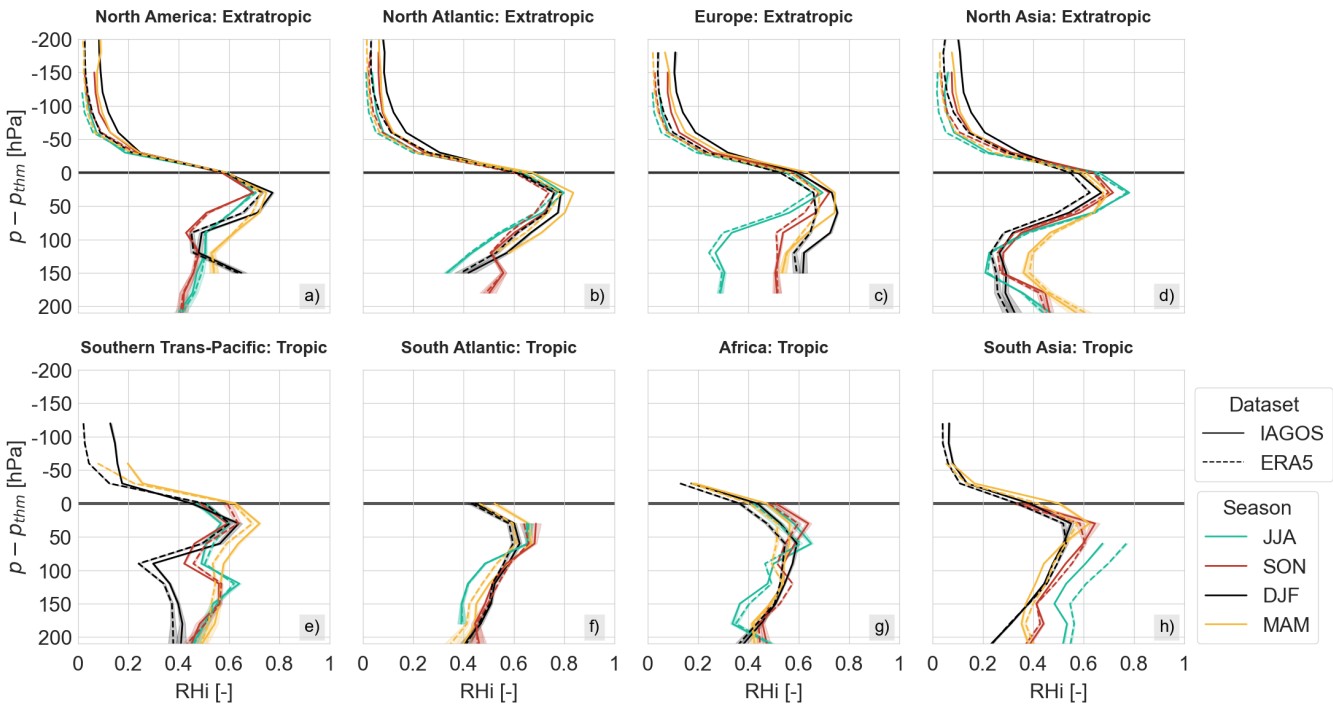

**Figure 9. (a-h)** Vertical distribution of IAGOS and ERA5 mean relative humidity over ice per season and per region with shading showing the 95% confidence interval, using levels based on distance to thermal tropopause, only considering levels, seasons and regions with 500+ samples.

The seasonal behaviour of RHi is more apparent in the extratropics compared to the tropics. In North America, North Atlantic, and Europe, the highest RHi is observed for seasons DJF and MAM, the seasons with the lowest mean temperatures as seen in Fig. 7. These are also the two seasons where we find a larger dry bias in ERA5. This is due to higher values of RHi favouring lower temperatures (Sanogo et al., 2024), and ERA5 shows more inconsistencies in the prediction of large

RHi as shown in Fig. 6 and by Wolf et al. (2025). It is also interesting to note that these differences appear smallest in North





America and increase as we move towards Europe, showing a possible longitudinal dependency. This was also observed by Wolf et al. (2025), who hypothesised it could due to the spatial distribution of water vapour. From a global distribution of the mean RHi and ISS occurrence using IAGOS, we see that North America is drier, with less ISS occurrences, compared to the North Atlantic and Europe, which could result in larger differences in the two latter regions due larger biases in ERA5 close
to and in ISS conditions (Petzold et al., 2020; Wolf et al., 2025). Just below the tropopause and in the lower stratosphere, the seasonality of RHi disappears, as also shown by Petzold et al. (2020).

While we do not observe a seasonal cycle in RHi in the tropics, we do find that South Asia has a larger mean RHi in season JJA compared to the other seasons in this region. The higher RHi may be the result of the tropics being a deep-convection region with strong updrafts, which result in ISS and high nucleation rates leading to a high number of ice particles (Sanogo et al.,
2024). Petzold et al. (2017) also found a correlation between a high number of ice particles and high values of RHi. In fact, we find a higher frequency of cloudy conditions for season JJA compared to other seasons in South Asia when observing IAGOS measurements, showing that more measurements with larger number of ice particles are found in this season. Interestingly, for season JJA in South Asia, we also identify a moist bias in ERA5. This is also the same region and season for which the minimum temperature did not coincide with the location of the thermal tropopause. Hence, we theorise that the moist bias may
also be related to the type of weather encountered in South Asia in season JJA.

Overall, the expected differences in RHi between IAGOS and ERA5 are not governed by the location relative to the local tropopause, but rather by the atmospheric layer, i.e. upper tropopause or lower stratosphere. The largest biases in ERA5 are expected in the upper troposphere, with a tendency for ERA5 to be drier than IAGOS. Moreover, in the extratropics, colder months tend to result in a larger dry bias in ERA5.

## 3.3 Distribution of relative humidity over ice from IAGOS and ERA5 under cloudy and clear-sky conditions

In this section, we investigate the effect of clear-sky, indeterminate, and cloudy conditions based on the probability density function of RHi from IAGOS and ERA5. Instead of discrete vertical levels, we consider the three atmospheric layers because of the low sampling of the number of ice particles in IAGOS. The three layers are upper troposphere (UT), tropopause (TROP) and lower stratosphere (LS). The UT is defined as $p_{thm} > 15$ hPa, the TROP as $-15$ hPa $\leq p_{thm} \leq 15$ hPa and LS as $p_{thm} < -15$
hPa.

Figure 10 and Fig. 11 display the probability density function (PDF) for the different cloudy conditions, per atmospheric layer, and per geographic region considered in this study. In the lower stratosphere, we mainly observe the clear-sky PDF of RHi due to rare cloud occurrence in this atmospheric layer. The clear-sky PDF is governed by low values of RHi and low probability of ISS, with a general monotonically decreasing behaviour. This was reported by Gierens et al. (1999) using the
global distribution of MOZAIC flights and it was also seen by Sanogo et al. (2024) for the lower stratosphere in high-latitude regions (60-80°N) using IAGOS measurements. In the tropic regions, the clear-sky PDF in the LS shows some multimodal behaviour, but this is most likely the result of a small number of samples in this atmospheric layer (see Fig. 2). Both the extratropics and tropics show small differences in low values of RHi in the LS. This is not necessarily due to biases in ERA5, but may be due to limitations of the IAGOS ICH sensor as discussed in Sect. 3.2. However, this will not impact the prediction



of ice supersaturated regions as the issue only arises below RHL ≈ 10%, which is equivalent to RHi ≈ 15-18% given the mean temperature in the LS.

The tropopause layer is not completely dry (Petzold et al., 2020; Reutter et al., 2020), hence we observe that ISS is possible in clear-sky, cloudy and indeterminate conditions, as shown in Fig. 10 and Fig. 11. As is evident from the comparison of the IAGOS and ERA5 PDF, ERA5 shows good approximation of RHi below ISS in clear-sky conditions until RHi ≈ 0.75 − 0.90,

with a lower probability of values greater than 1 compared to IAGOS. The underestimation of ISS in ERA5 for clear-sky conditions may be the result of the resolution of ERA5, which only provides hourly mean and grid cell values (Schumann et al., 2021). However, it shows that by lowering the threshold value of RHi for ISS, we could artificially increase the prediction of ISS in ERA5, though the value may differ per region; as seen from Fig. 10, the RHi threshold value would be close to 1 for North America, but it might be closer to 0.75 in Europe. On the other hand, indeterminate and cloudy conditions are governed

by peaks at RHi = 1 for ERA5, which is due to saturation adjustment. For IAGOS, we also observe peaks for these conditions, but their distributions are wider and centred at higher values of RHi, as cirrus clouds often exhibit an ice supersaturated wet mode, but they can also be subsaturated (Sanogo et al., 2024). Similar behaviour was observed by Wolf et al. (2025).

The upper troposphere has more moisture compared to the tropopause and lower stratosphere, allowing for higher probabilities of ISS occurrence. For clear-sky conditions, there is a smaller probability of low RHi, with increased probability of

higher RHi, and a more equal probability across all observed RHi values. ERA5 shows a drop-off in probability just before RHi = 1, as was also seen in the tropopause. Again, this underestimation in ISS occurrence could be corrected by lowering the RHi threshold value for ISS in ERA5. Cloudy and indeterminate conditions in ERA5 for the UT are also governed by peaks centred around RHi = 1, but with higher probabilities due to larger occurrences of these two conditions than at the TROP (see Fig. 4). As seen at the TROP, IAGOS is centered at values of RHi just above 1 in the UT for these two conditions, with higher

probabilities. Again, the overall behaviour is similar to that found by Wolf et al. (2025), but we find larger difference in the peak probability between IAGOS and ERA5 for cloudy and indeterminate conditions, which may be the result of taking into account the different atmospheric layers. If we consider the same geographic area, pressure levels and time frame for IAGOS measurements as Wolf et al. (2025) and do not consider separate atmospheric layers and do not normalise given the number of observations, we obtain more similar results to Wolf et al. (2025). This shows that the atmospheric layer plays a role in the

behaviour of the PDF.

Comparison of extratropic and tropic PDFs for cloudy and indeterminate conditions, shows similar behaviour. However, in South Asia, this mode is located at values lower than 1, i.e. it is subsaturated. Sanogo et al. (2024) found similar observations when considering a larger tropical region. While we do not find this observation in the other tropic regions, it makes sense that Sanogo et al. (2024) finds this overall observation due to the high percentage of measurements in the South Asia region. This

observation in South Asia may also provide insights into the moist bias found in Sect. 3.2. The mean vertical distribution of RHi shows a higher mean RHi for ERA5 compared to IAGOS in cloudy and indeterminate conditions for South Asia in JJA, at vertical levels below the tropopause. This is because the mean RHi in IAGOS for this particular region and season is less than 1 in cloudy and indeterminate conditions, which is inline with the behaviour of the PDF, whereas ERA5 shows a mean of 1 due to the saturation adjustment. Therefore, the moist bias appears to arise as a result of the saturation adjustment, which





cannot take into account that the (wet) mode of the RHi PDF can be subsaturated in some regions. Whether the subsaturation is a result of the specific season in South Asia is uncertain and requires further evaluation.

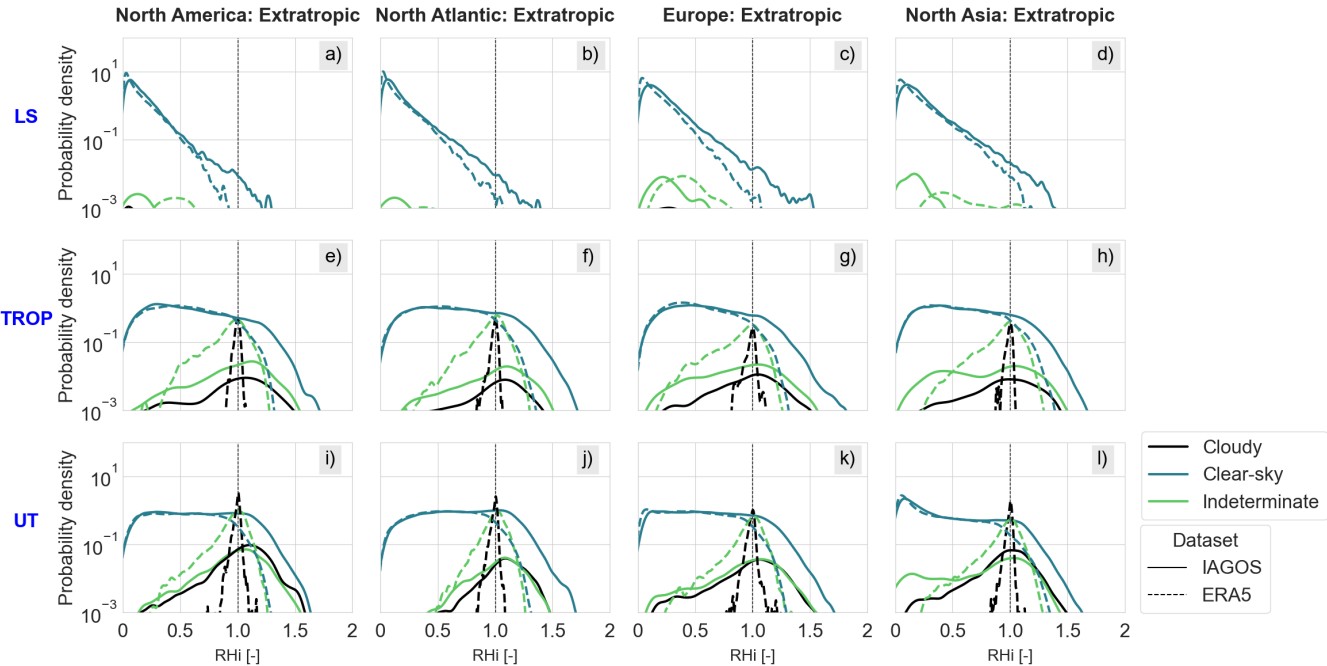

**Figure 10. (a-l)** Probability density function of IAGOS and ERA5 relative humidity over ice in the upper troposphere (UT), at the tropopause (TROP) and in the lower stratosphere (LS) for cloudy, clear-sky and indeterminate conditions in the four extratropic regions. The PDFs per subplot are normalised with respect to the number of observations within each subset of IAGOS or ERA5 used for that subplot.





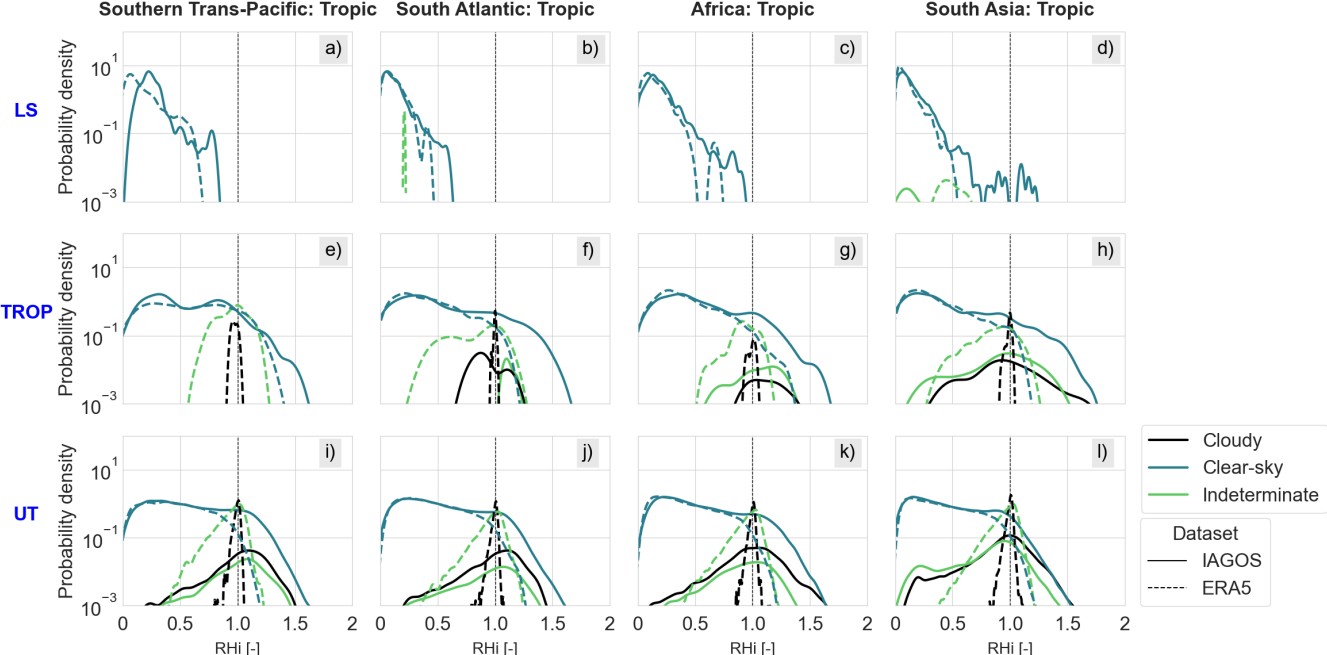

**Figure 11. (a-l)** Probability density function of IAGOS and ERA5 relative humidity over ice in the upper troposphere (UT), at the tropopause (TROP) and in the lower stratosphere (LS) for cloudy, clear-sky and indeterminate conditions in the four tropic regions. The PDFs per subplot are normalised with respect to the number of observations within each subset of IAGOS or ERA5 used for that subplot.

Overall, ERA5 tends to estimate RHi well in clear-sky conditions, until just before ISS, but underestimates RHi in ISS conditions. Hence, for clear-sky conditions, we can expect an underestimation of ISSRs, but this could be improved by lowering the threshold value of RHi for ISS. However, in cloudy and indeterminate conditions, ERA5 shows narrow RHi distributions
centred at 1 due to the saturation adjustment, while IAGOS shows that higher values can occur in such conditions. As a result of this, ERA5 may predict less ISSRs compared to ERA5. IAGOS also shows that some regions, such as South Asia, can have a subsaturated wet mode in cloudy and indeterminate conditions that ERA5 cannot predict due to the saturation adjustment. This may lead to an overestimation of ISSRs in ERA5.

### 3.4    Distribution of ice supersaturated regions in IAGOS and ERA5

Comparison of the RHi showed a general dry bias in ERA5, compared to IAGOS, with some seasonal and regional differences. In the following section, we will explore the impact of such biases on the ice supersaturated region occurrence.

Figure 12 displays the vertical distribution of the ISSR fraction per season and geographical region. The ISSR fraction is calculated by finding the total number of points showing ISS conditions and dividing by the total number of points, per vertical level, season and geographical region. There is an overall underestimation of ISSRs in ERA5 due to the dry bias in RHi.
This was also reported by Reutter et al. (2020) when considering the North Atlantic region. In instances where the RHi is overestimated by ERA5, such as in South Asia for season JJA, the ISSR fraction is overestimated by ERA5. Agarwal et al.





(2022) finds a tendency for ERA5 to overestimate the ISSR occurrence at cruise altitudes in the tropics and mid-latitudes when compared to radiosonde measurements. However, the radiosonde measurements are not corrected for dry biases and the location of these radiosonde observations are generally over land and not very widespread as with the IAGOS measurements,

which could lead to differences in obtained results.

From a seasonal perspective, there is a clear seasonal pattern in North America, North Atlantic and Europe, with seasons DJF and MAM showing highest ISSR occurrence. This is in line with other research, such as done by Petzold et al. (2020) and Sanogo et al. (2024). However, we do not find a distinct seasonal behaviour in the tropics, just as with the relative humidity over ice. This is also in line with Sanogo et al. (2024), who found the ISSR frequency to vary by around 5% or less among the

four considered seasons in the tropics.

We find that the maximum ISSR fraction in the extratropic regions is between 30 and 35% for seasons DJF or MAM using IAGOS measurements. The maximum ISSR fraction occurs just below the tropopause, around 30 hPa (approximately 750 m at cruise level) (Thouret et al., 2006; Petzold et al., 2020) and reduces to zero in the lower stratosphere. This is similar to what was reported by Petzold et al. (2020). On the other hand, Reutter et al. (2020) found a maximum of 40% using IAGOS when

considering all seasons, but with the maximum also occurring 30 hPa below the tropopause and a reduction of the ISSR fraction to zero in the lower stratosphere. Sanogo et al. (2024) found a maximum ISSR frequency of around 20% in the mid-latitudes for seasons MAM and DJF. However, the ISSR fraction is highly sensitive to the number of points available and these studies use a different subset of the IAGOS/MOZAIC dataset. In the tropics, we report an increasing ISSR fraction until just below the tropopause, with a maximum of approximately 20%, but is independent of the season. Sanogo et al. (2024) reports a maximum

ISSR fraction of 15% in the tropics using IAGOS and with no seasonal variation. The ISSR fraction also increases with altitude for pressure levels below the average thermal tropopause height in the tropics. Lamquin et al. (2012) also showed increasing ISSR occurrence with altitude in the tropics using MOZAIC, with maximum ISSR frequencies of up to 30%, but this study considered the frequency per grid box and not an overall defined region.





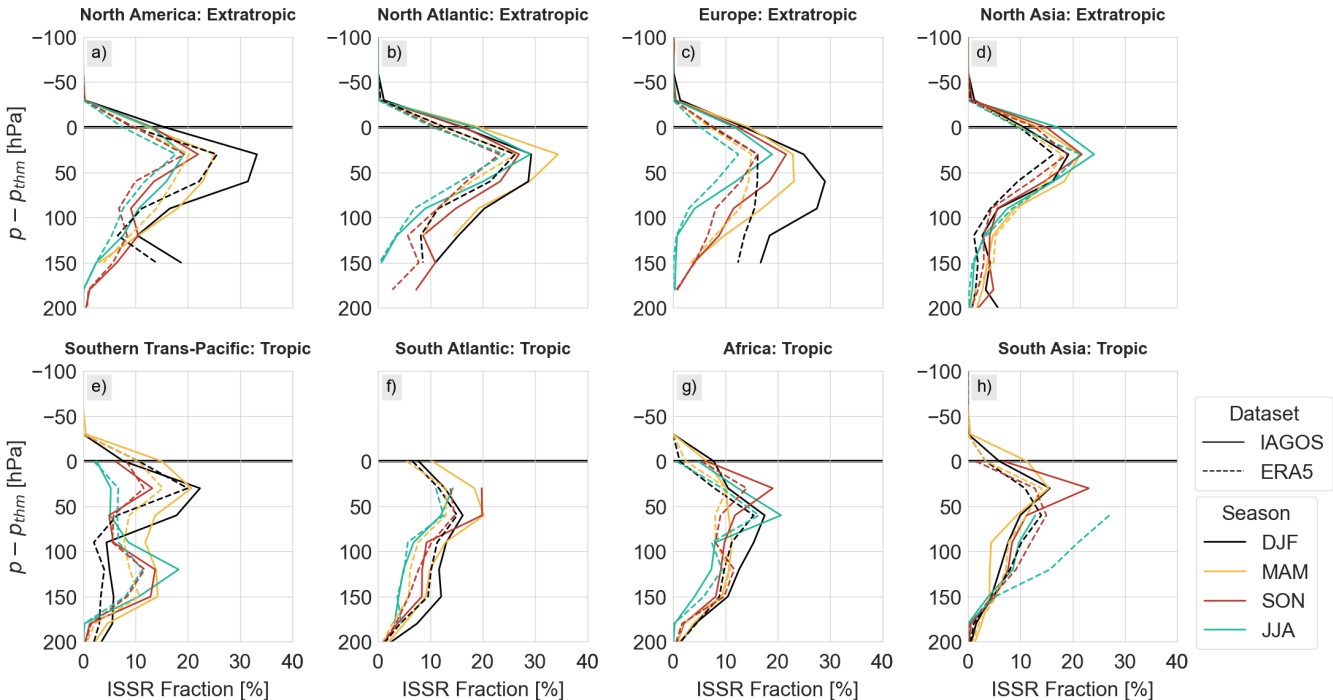

**Figure 12. (a-h)** Vertical distribution of IAGOS and ERA5 ice supersaturated region fraction per season and geographical region. Only considered levels, seasons and regions with 500+ samples.

To quantify the skill of ERA5 in predicting ISSRs, we can use the equitable threat score (ETS). This performance measure is preferred over measures such as the hit rate and false alarm rate since ISSRs can be a rare event (Gierens et al., 2020a; Wolf et al., 2025), for example in the lower stratosphere. The ETS is calculated using Eq. 1, where $r$ is found with Eq. 2 and the other variables are entries in the contingency table shown in Table 4. If ETS = 1, it means that there is a perfect correlation between the observation and the prediction (Gierens et al., 2020a). When ETS = 0, it means that the relationship is completely random (Gierens et al., 2020a).

$$\text{ETS} = \frac{\text{TP} - r}{\text{TP} + \text{FN} + \text{FP} - r} \tag{1}$$

$$r = \frac{(\text{TP} + \text{FN})(\text{TP} + \text{FP})}{\text{TP} + \text{FN} + \text{FP} + \text{TN}} \tag{2}$$



**Table 4.** Contingency table definition

| IAGOS observation | ERA5 prediction | |
| --- | --- | --- |
| | Yes | No |
| Yes | True positive (TP) | False negative (FN) |
| No | False positive (FP) | True negative (TN) |

Figure 13 shows the vertical distribution of the ETS per season and geographical region. Overall, the ETS lies between 0.2 and 0.4 in the upper troposphere, depending on the season, vertical distance to the tropopause and region. This means that there is a weak to mediocre correlation between IAGOS and ERA5 in the prediction of ISSRs. The high ETS seen for North Atlantic

in season DJF at 150 hPa below the tropopause and for North America in season JJA at 180 hPa below the tropopause appear to be outliers, even though more than 500 samples have been used to calculate the ETS at these conditions (see Sect. S1 in the Supplement). Otherwise, the ETS remains relatively constant with distance from the tropopause in the upper troposphere. It begins to decrease at the tropopause and reduces further in the lower stratosphere. We also find low ETS at 200 hPa below the tropopause. Both in the lower stratosphere and well below the tropopause, ISSR occurrence is rare (see Fig. 12). Hence, ERA5

shows decreased ISSR predictive skills when its occurrence is rare. When the ETS is 0, such as 60 hPa above the tropopause in North America, it is the result of IAGOS not observing any ISSRs, but ERA5 predicting their occurrence, which results in ERA5 showing no skill. Gierens et al. (2020a) calculated ETS between 0.05 and 0.25, depending on the season. The value of 0.05 is on the lower side, but this may be because the study does not take into account the different atmospheric layers or regions, and also only considers four months of IAGOS measurements. If we do not take into account the vertical layers

defined, the ETS remains between 0.15 and 0.3. Hence, the low ETS found by Gierens et al. (2020a) is most likely the result of using a smaller subset of IAGOS and ERA5 data. The study by Wang et al. (2025) shows an ETS of 0.23 upper troposphere (UT) and 0.14 in the lower stratosphere before application of the machine learning algorithm, which agrees well with the values identified in this study.

    From a regional perspective, we find a higher ETS in North America, which decreases as we move towards Europe. For

these three regions, we also found a possible longitudinal dependency in the RHi difference, discussed in Sect. 3.2, where North America has the lowest difference and Europe the highest. Hence, the lower RHi difference leads to a higher ETS, showing better ISSR prediction skills in ERA5. In the tropics, the ETS is lower than in the extratropics at most vertical levels defined. At 30 hPa below the tropopause, the ETS increases, but the reason is unknown given that we tend find larger differences in RHi for these distances from the tropopause. On the other hand, South Asia in season JJA has one of the lowest ETS, which

is the result of the moist bias seen in Sect. 3.2, leading to more FPs in the prediction of ISSRs.

    There is some seasonal variation in the ETS; the maximum difference between seasons is approximately 10%, at most. In North America, we find that the variation can be up to 20%. Nevertheless, in the extratropics, we tend to find that season SON and DJF results in the highest ETS and season JJA and MAM in the lowest. Gierens et al. (2020a) also showed approximately 10% variation in the ETS between seasons, with the largest ETS occurring in January (winter), equivalent to DJF, and lowest




ETS in July (summer), equivalent to JJA. Hence, this shows that in the extratropics, we tend find a better agreement in the prediction of ISSRs between IAGOS and ERA5 for fall and winter. No specific seasonal trends are identified for the ETS in the tropics.

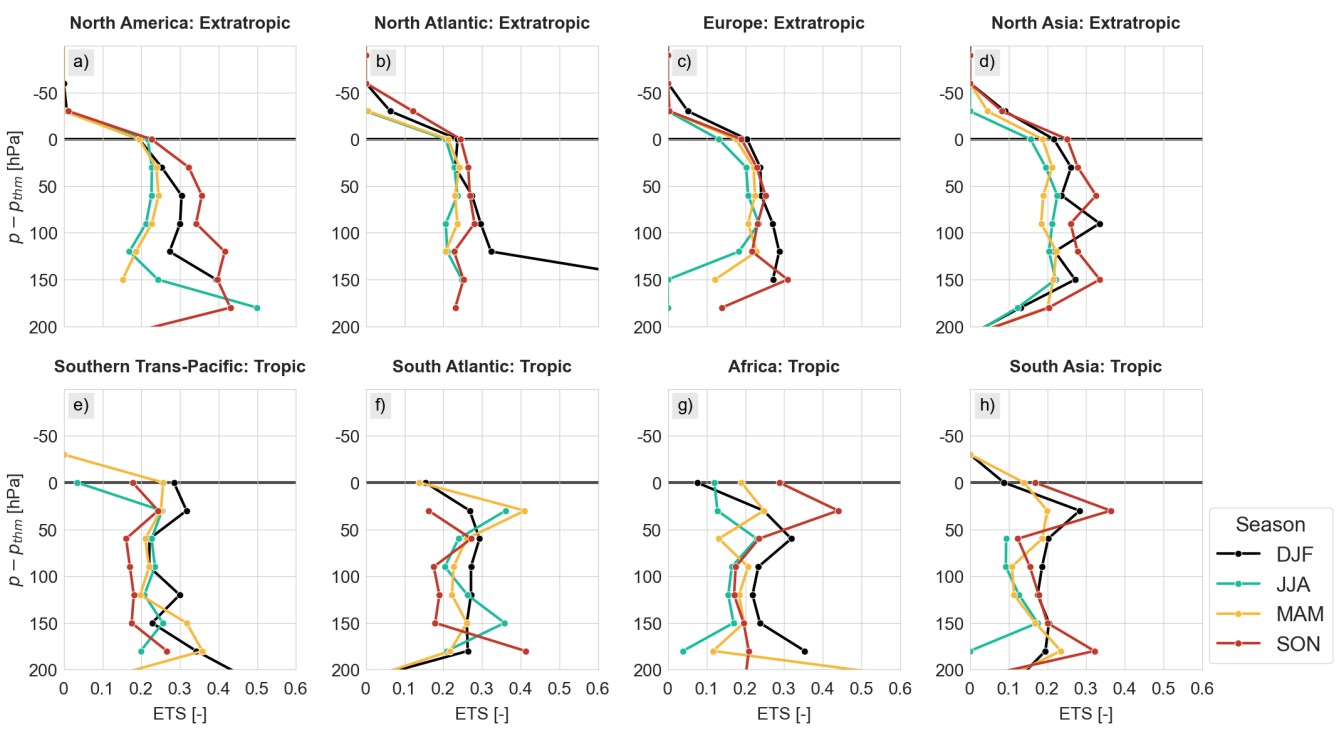

**Figure 13. (a-h)** Vertical distribution of ice supersaturated region equitable threat score per season and geographical region for ERA5. ETS is calculated for level, season and region for which there are 500+ samples.

### 3.5 Prediction of ice supersaturated regions in ERA5 under cloudy and clear-sky conditions

In Sect. 3.3, the effect of cloudy conditions on the RHi was investigated for the different atmospheric layers and for the different
geographical regions. Here, we explore the effect of cloudy and clear-sky conditions on the capability of ERA5 to correctly identify ice supersaturated regions. Again, we only consider the three atmospheric layers due to low sampling of the number of ice particles in IAGOS, as described in Sect. 3.3.

Table 5 displays the ETS of cloudy, clear-sky and indeterminate conditions for the different geographic regions in the different atmospheric layers. Note that for this analysis we consider conditions with 250+ samples instead of 500+ samples,
as done previously, due to less measurements with number of ice particles within IAGOS. However, as seen in Sect. S1 in the Supplement, 250+ samples should still be sufficient for the calculation of the ETS.

The main observation is that the ETS is highest in clear-sky conditions and lowest in cloudy and indeterminate conditions in the extratropics. In clear-sky conditions, we find an ETS of approximately 0.15 in the extratropic UT and TP. This indicates a





weak coherence in the prediction of ISSRs between IAGOS and ERA5. The ETS reduces to between 0 and 0.05 in the clear-sky
LS due to the rare occurrence of ISSRs and the underestimation of ISSRs by ERA5. For indeterminate conditions, the ETS
reduce to less than 0.1 in the UT and TP, except in North Asia, where it ranges between 0.11 and 0.14. For cloudy conditions,
the ETS is between 0.05 and 0.08 in most extratropic regions, indicating an almost purely random relationship between IAGOS
and ERA5. In North America, we find the ETS in the cloudy UT to be 0.14, showing a more weak coherence. No ETS are
calculated in the extratropic TP or LS due to insufficient samples. This means that ERA5 shows better skill at predicting ISSR
conditions under clear-sky conditions, albeit it is a weak relationship.

The weak coherence in the extratropic UT under clear-sky conditions is most likely the result of the clear-sky dry bias in
ERA5, discussed in Sect. 3.3. Teoh et al. (2022) found that the ETS could be improved by lowering the RHi threshold for ISS,
which we also discussed in Sect. 3.3, where the PDFs could be used to identify the necessary threshold to apply for each region.
For cloudy and indeterminate conditions, the almost random relationship may be the result of the saturation adjustment in the
IFS, given that we are matching the clear-sky, cloudy and indeterminate conditions between IAGOS and ERA5. This means
that ERA5 cannot accurately predict the value of RHi within a cirrus cloud and may underestimate or overestimate ISSRs as a
result.

In the tropics, the ETS is generally 0.05 in the clear-sky UT and show little increase or decrease in the TP. The ETS is 0
in the LS due to no ISSRs. In the cloudy and indeterminate UT, the tropics shows an ETS of 0.1 or less, showing an almost
entirely random relationship. The lower ETS in the tropics is also inline with the results presented in Sect. 3.4 and shows that
ERA5 has little to no skill in predicting ISSR occurrence in tropic regions.

**Table 5.** Equitable threat score from prediction of ISSRs under clear-sky, cloudy and indeterminate conditions in ERA5 in upper troposphere (UT), tropopause (TP) and the lower stratosphere (LS). The different conditions have been matched between IAGOS and ERA5. Only combinations with 250+ samples are considered.

| | Clear-sky | | | Cloudy | | | Indeterminate | | |
|---|---|---|---|---|---|---|---|---|---|
| | UT | TP | LS | UT | TP | LS | UT | TP | LS |
| **North America** | 0.13 | 0.15 | 0 | 0.14 | - | - | 0.09 | 0.08 | - |
| **North Atlantic** | 0.16 | 0.16 | 0.05 | 0.05 | - | - | 0.06 | 0.03 | - |
| **Europe** | 0.15 | 0.13 | 0.05 | 0.08 | - | - | 0.07 | 0.03 | - |
| **North Asia** | 0.18 | 0.18 | 0.06 | 0.07 | - | - | 0.14 | 0.11 | - |
| **Southern Trans-Pacific** | 0.05 | 0.09 | 0 | - | - | - | 0.1 | - | - |
| **South Atlantic** | 0.1 | 0.03 | 0 | 0.06 | - | - | 0.08 | - | - |
| **Africa** | 0.05 | 0.08 | 0 | 0.01 | - | - | 0.04 | - | - |
| **South Asia** | 0.05 | 0.04 | 0 | 0.08 | - | - | 0.05 | - | - |





Wang et al. (2025) found an ETS score of 0.06 in clear-sky upper troposphere and lower stratosphere (UTLS). In the cloudy UTLS, Wang et al. (2025) calculated an ETS score of 0.23. These different ETS scores are most likely the result of using different methodologies for classifying cloudy and clear-sky conditions. Wang et al. (2025) uses the specific cloud ice water content from ERA5 to determine if a point is within a cloud or in clear-sky. Other papers use the number of ice particles to determine cloudy conditions in IAGOS measurements (Petzold et al., 2017; Sanogo et al., 2024; Wolf et al., 2025), but the reporting of cloudy conditions compared to ERA5 appears relatively new. Only Wolf et al. (2025) and Wang et al. (2025) seem to have considered such a comparison, but use different variables to determine cloudiness in ERA5. Hence, it raises the question for the correct procedure for classifying a point as cloudy or clear-sky.

## 3.6 Prediction of ice supersaturated regions in ERA5 for different North Atlantic weather patterns

As discussed by Irvine et al. (2012), there is a dependency of ice supersaturated region occurrence on different North Atlantic weather patterns. In this section, we explore if they also have an impact on the biases to expect in ERA5. We consider eastbound and westbound routes separately as eastbound routes tend to take advantage of the jet stream to reduce fuel consumption.

Figure 14 shows the ETS for the winter and summer weather patterns on the eastbound routes. The summer weather patterns appear to have no effect on the ETS, however, these patterns are also weaker than the winter weather patterns due to weaker teleconnection patterns and less variation of the jet stream latitude (Irvine et al., 2013). For the winter weather patterns, we find a tendency for W1, W2 and W4 to have a lower ETS compared to W3 and W5, except at 60 hPa below the tropopause and in the lower stratosphere. In the lower stratosphere, all winter weather patterns have almost equal ETS, except for W2, which has a higher ETS. Irvine et al. (2012) found that W2 showed a higher frequency of ISSRs at higher altitudes when using ERA-Interim. We find that ERA5 shows a slightly higher ISSR fraction in the lower stratosphere compared to IAGOS, but the difference is less than 1% and can therefore not be considered significant. Hence, the larger ETS for W2 in the lower stratosphere most likely arises from low ISSR occurrence in combination with the slightly higher ISSR fraction in ERA5. Between 30 hPa from the tropopause and the tropopause itself, W1, W2 and W4 have an ETS between 0.15 and 0.2, showing a weak coherence between IAGOS and ERA5 in the prediction of ISSRs. For W3 and W5, the ETS improves to the range 0.2 to 0.25, but this is still a weak to mediocre coherence. The main difference between these two groups of weather patterns is their jet stream strength. That is, W1, W2 and W4 are classified as having a strong jet stream and W3 and W5 have a weak jet stream.

The question that arises is how the characteristics of the different weather patterns affect ISSR occurrence. For example, Irvine et al. (2012) showed that for W4, the ridge over the Atlantic is most pronounced, which leads to high frequency of ISSRs over Greenland and low ISSR occurrence south of the jet stream, when using ERA-Interim. Eastbound flights fly at these more southern latitudes (Irvine et al., 2012). Due to ISSRs being a more rare occurrence along eastbound routes for W4, also seen in the IAGOS ISSR fraction, it could lead to ERA5 predicting less of the ISSRs that do occur. We also find a lower ISSR fractions for W1 and W2 using IAGOS. Hence, we see a tendency for lower ISSR occurrence in IAGOS leading to a lower ETS, indicating a lower predictive skill for ERA5. Whether the jet stream strength or jet stream position are the reason for the changes in the ISSR occurrence and how it can impact the ability of ERA5 to predict ISSRs requires further research.



Irvine et al. (2012) did find changes in the ISSR frequency between winter weather patterns, but this study only considered ERA-Interim, which is also known to have issues in the prediction of ISSRs (Reutter et al., 2020).

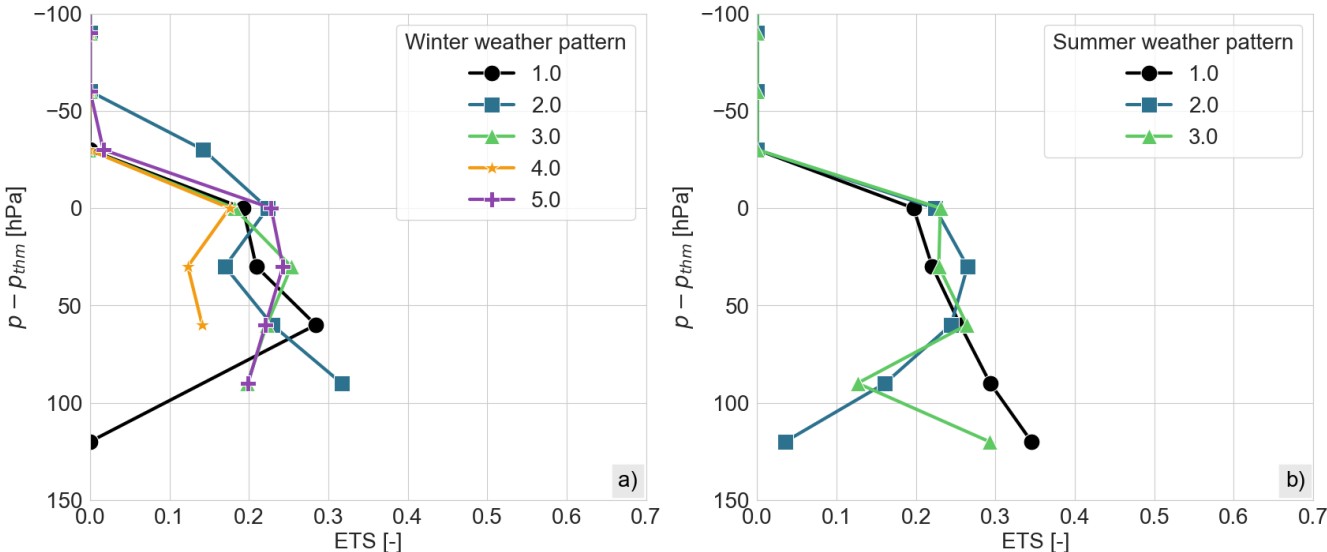

**Figure 14.** Vertical distribution of ETS from prediction of ISSRs in ERA5 on eastbound routes over the North Atlantic for **a)** winter and **b)** summer weather patterns. Only combinations of weather pattern and vertical level with 250+ samples are considered.

Figure 15 shows the vertical distribution of the ETS on westbound routes for the winter and summer weather patterns. Again, no significant differences are found between the summer weather patterns. For the winter weather patterns, we do not find large
differences in the ETS, except near or at the tropopause, or at a distance of 100 hPa or more below the tropopause. Irvine et al. (2012) also found smaller differences in the probability of forming contrails between each winter weather type for westbound time-optimal routes, which were also lower than eastbound time-optimal routes. We also generally find lower ISSR fractions for each winter weather type on westbound routes compared to eastbound routes, except for W4. This could be related to the higher ISSR occurrence at more northern latitudes compared to southern latitudes for W4 (Irvine et al., 2012), where we also
find the mean latitude of westbound IAGOS flights to be more north compared to eastern bound flights.

While the ISSR fraction tends to be lower for western bound flights, the ETS is shifted to slightly higher values. This shows a somewhat better agreement between IAGOS and ERA5 on westbound routes. We theorise that it could be related to the distribution of ISSRs that result from each winter weather pattern type. For example, the distribution of ISSRs may be more patchy close to and along the jet stream, but larger ISSRs are found further from the jet stream, which ERA5 may be better able
to predict. In fact, Irvine et al. (2012) showed different distributions and frequencies of ISSRs in ERA-Interim for the different winter weather patterns, where a less patchy area of high ISSR frequency was found over Greenland compared to along the jet stream. However, this was averaged over several winters and thus, we cannot identify if the larger areas are due to averaging or if there exists larger areas of ISSRs at more northern latitudes. To understand if the distribution of ISSRs under different winter



weather patterns impact the ability of ERA5 to predict ISSRs, it would be necessary to analyse the daily variability of ISSRs
as the jet stream position and strength can show variation within the weather pattern itself.

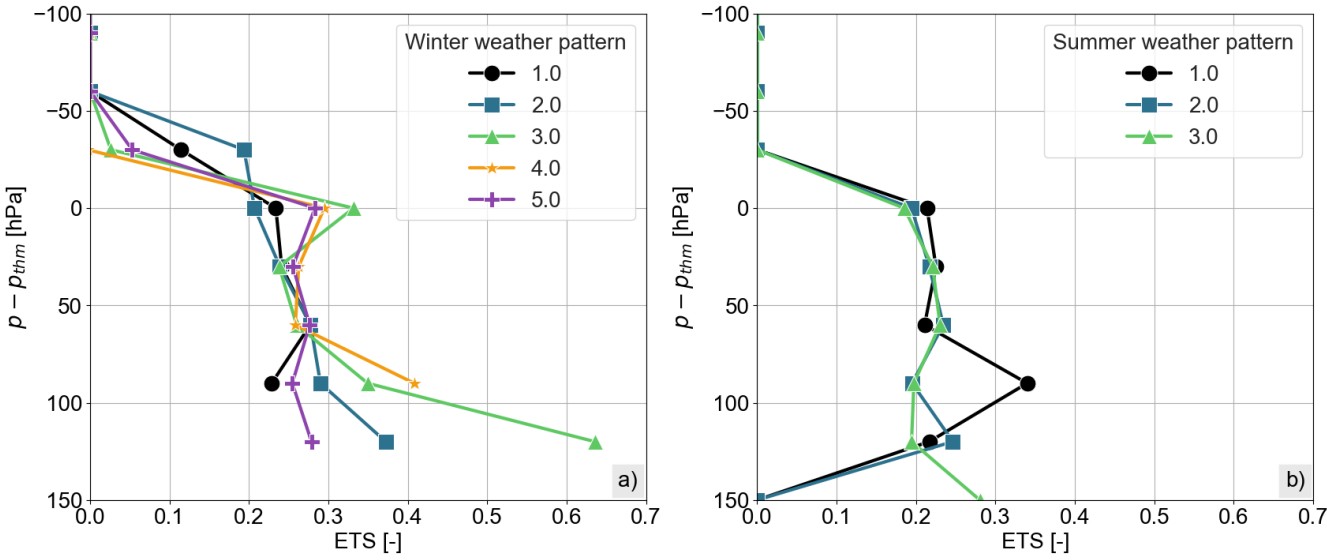

**Figure 15.** Vertical distribution of ETS from prediction of ISSRs in ERA5 on westbound routes over the North Atlantic for **a)** winter and **b)**
summer weather patterns. Only combinations of weather pattern and vertical level with 250+ samples are considered.

## 4    Conclusions

In this study, we evaluated ERA5 temperature, relative humidity over ice (RHi) and ice supersaturated region (ISSR) occurrence
against IAGOS in situ measurements over the period 2011-2022. Differences in the frequency of ice supersaturation (ISS) were
also compared. The analysis was performed over a large geographical area covering the tropics and extratropics. It included the
upper troposphere (UT) and lower stratosphere (LS), with vertical levels defined based on distance to the thermal tropopause.
The RHi and ISSR occurrence is also documented for cloudy and clear-sky conditions. We used the number of ice particles in
the IAGOS measurements and the cloud cover in ERA5 to distinguish between cloudy, clear-sky and indeterminate conditions.
The impact of different North Atlantic weather patterns on ERA5s ability to predict ISSRs was also investigated. The main
conclusions of our study are as follows:

1. The vertical distribution of temperature and relative humidity over ice was analysed per season and region for IAGOS
    and ERA5. Temperature differences between IAGOS and ERA5 were mainly a function of the atmospheric layer, with a
    larger cold bias in the LS. ERA5 showed a dry bias in RHi for all atmospheric layers, but the lower stratospheric biases
    may be due to limitations of the IAGOS ICH sensor. Larger dry biases were found in colder months in the extratropics
    due to larger values of RHi being possible in these conditions. A moist bias was identified in South Asia for season JJA.



2. We characterised the PDF of RHi in the UT, the tropopause (TROP) and the LS for cloudy, clear-sky and indeterminate conditions. In clear-sky conditions, ERA5 showed an underestimation of ISS. In cloudy conditions, ERA5 displayed large peaks centred at RHi $= 1$, due to the scheme that lowers RHI to saturation when clouds are present (saturation adjustment). This is more critical in the UT and TROP as the LS is dry and cloud occurrence is rare, with little to no ISS conditions. While this generally also results in a dry bias, it can also cause a moist bias if the wet mode of the RHi PDF is subsaturated.

3. ERA5 shows a general underestimation of ISSRs in comparison to IAGOS due to biases in the RHi. The ability of ERA5 to predict ISSRs was estimated using the equitable threat score (ETS). It showed a weak to mediocre correlation between IAGOS and ERA5 in the prediction of ISSRs, with a weaker coherence in the tropics compared to the extratropics. ERA5 also showed a better ETS in fall and winter compared to summer and spring in the extratropics, but no seasonal dependence was identified in the tropics. The lowest ETS was found for South Asia in season JJA, which is most likely the result of ERA5 not being able to predict the subsaturated wet mode of the RHi PDF due to the saturation adjustment. For this region and season, the ISSR fraction was also overestimated by ERA5 in comparison to IAGOS.

4. The ETS was also calculated for the prediction of ISSRs under cloudy, clear-sky and indeterminate conditions. It showed a better coherence between IAGOS and ERA5 for clear-sky conditions compared to cloudy conditions, particularly in the extratropics. The ETS values for cloudy conditions indicate an almost random relationship, showing ERA5 cannot accurately predict ISSRs under such conditions. The tropics showed almost random relationships for both cloudy and clear-sky conditions.

5. The influence of the North Atlantic weather patterns on ERA5s capability to predict ISSRs was also investigated using the ETS. The winter weather patterns resulted in stronger differences between each type of pattern compared to summer weather patterns. This is the result of weaker summer patterns. For the winter weather pattern, we see more distinct differences between each weather type on eastbound routes compared to on westbound routes. We theorise it could be due to the distribution of ISSRs due to the jet stream, but it requires further investigation.

This study is an extension of previous studies comparing IAGOS and ERA5, such as Gierens et al. (2020a) and Wolf et al. (2025). These studies also find ERA5 to underestimate ISSR occurrence compared to in-situ measurements due to biases in estimation of RHi in ERA5. However, they are limited in their regional and seasonal coverage. Our analysis considers geographical areas not covered in the previously mentioned studies, such as the tropics, and provides a seasonal comparison within each subregion considered. It shows that we cannot assume that ERA5 has the same ability to predict ISSRs in different subregions and that ERA5 has less ability to predict ISSRs in the tropics. Our seasonal comparison reveals a similar seasonal pattern in ISSR occurrence in the extratropics as reported in previous studies, but we further find that the ability of ERA5 to predict ISSRs can itself seasonally dependent.

With regard to the comparison of ice supersaturation in cloudy, clear-sky and indeterminate conditions, we find comparable results in the RHi PDF in the extratropics to those of Wolf et al. (2025), and the IAGOS RHi PDF closely resembled the



results obtained by previous studies. We find larger differences in the peak probability of RHi between IAGOS and ERA5 under cloudy and indeterminate compared to Wolf et al. (2025), most likely due to the consideration of the atmospheric layers in this study. Moreover, we extended the analysis to the tropics and also quantified how cloudy, clear-sky and indeterminate conditions affect the ability of ERA5 to predict ISSRs, which was shown to be driven by the clear-sky dry bias in RHi and the saturation adjustment in ERA5. Wang et al. (2025) calculated the ETS for RHi > 100% with IAGOS and ERA5 under cloudy and clear-sky conditions, for which a higher ETS was found for cloudy conditions compared to clear-sky for a a dataset covering the upper troposphere and lower stratosphere. Wang et al. (2025) used a different methodology to detect clear-sky and cloudy conditions compared to this study. Thus, we recommend to adapt a standard methodology for comparing IAGOS cloudy and clear-sky conditions to cloudy and clear-sky conditions in ERA5.

The analysis is highly dependent on the number of available measurements from IAGOS and without enough measurements, our conclusions may not be representative. While we did evaluate the necessary number of points needed to be confident in our results and presented results with respect to these findings, there are still some limitations. In our evaluation on the prediction of ISSRs in ERA5 under cloudy and clear-sky conditions, we lowered the minimum sample size required for a representative result due to low sampling of the number of ice particles in IAGOS. This results in a larger variability of the ETS, and thus the true value may be higher or lower than reported, though, the 95% confidence interval lies close to the mean of 100 tests. Nevertheless, for future work, we recommended to increase the number of IAGOS aircraft capable of measuring the number of ice particles to further distinguish between cloudy and clear-sky conditions. For the North Atlantic weather pattern analysis, we also lowered the minimum sample size to better evaluate differences in the vertical distribution, which may have similar implications. Thus, we recommend to expand the analysis with more measurements in the future, if and when available. While the evaluation of the North Atlantic weather patterns on the ability of ERA5 to predict ISSRs indicated a possible dependence on winter weather patterns, the underlying cause remains uncertain. Hence, the effect of jet stream strength and position on ISSR occurrence and how it impacts the ability of ERA5 to predict ISSRs should be further researched and quantified before making definitive conclusions.

Based on the results of this study, the lack of ISS in clear-sky conditions is a key limitation in the ERA5 reanalysis dataset and should be addressed. Otherwise, ISSRs are systematically underestimated, potentially leading to an underestimation of the occurrence of persistent contrails and consequently, uncertainties in their estimated climate impact. The lack of ISS may temporarily be improved by decreasing the RHi threshold for ISS, but is only applicable to regions where we have in-situ measurements, limiting their global applicability. The analysis also shows that the application of the saturation adjustment in the NWP model underlying the reanalysis should be revisited to properly account for ISS under cloudy conditions as it contributes to inaccuracies in the prediction of ISSRs in ERA5. The influence of the jet stream strength and position on the distribution of ISSRs and the ability of ERA5 to predict ISSRs may also provide further insight into how ISSRs, and thus persistent contrails, could be avoided based on the weather pattern. Overall, these findings improve our understanding regarding the variability of ISSRs and the extent to which ERA5 is able to accurately predict ISSRs under different conditions. This has important implications for the accurate assessment of aviation induced persistent contrail formation, which can have a significant effect on the local radiative budget.



*Code availability.* The Python code that was used to perform the analysis is available upon request from the corresponding author.

*Data availability.* The IAGOS data can be obtained from the IAGOS data portal at https://doi.org/10.25326/20. The Analysis-Ready, Cloud
Optimized (ARCO) ERA5 data can be obtained from Google Cloud Storage at https://console.cloud.google.com/marketplace/product/
bigquery-public-data/arco-era5. The ERA5 data can be obtained from the ECMWF data catalog at https://doi.org/10.24381/cds.f17050d7
(Hersbach et al., 2023b).

## Appendix A:  Distribution of temperature and relative humidity over ice with respect to thermal and dynamic tropopause

In this appendix, we present the vertical distribution of temperature and relative humidity over ice with respect to the thermal
and dynamic tropopause. The tropopause data for both definitions are extracted from Hoffmann and Spang (2022). Since there
can be two thermal tropopauses (Hoffmann and Spang, 2022), we only consider the first thermal tropopause found.

Figure A1 shows the vertical distribution of temperature and relative humidity over ice for the thermal and dynamic
tropopause definitions. We find similar distributions with both definitions, though there are some differences. For the verti-
cal distribution of temperature, the dynamical and thermal tropopause shows similar values for the mean and the 25% and
75% percentiles, especially in the upper troposphere. In the lower stratosphere, it is noticeable that the temperature gradient
is sharper across the thermal tropopause in comparison to the dynamic tropopause. For the 99% percentile, we notice that the
thermal tropopause also has a sharper gradient in the upper troposphere, but for the 1% percentile it is stronger for the dynamic
tropopause. This is similar to observations by Petzold et al. (2020).

Larger differences between the two tropopause definitions are noticeable for the vertical distribution of relative humidity
over ice. In the lower stratosphere, larger values of RHi are found when using the dynamic tropopause definition, due to
the larger 99% percentile range, for which the thermal tropopause definition shows a sharper vertical gradient. This is also
shown by the smaller mean RHi for the thermal tropopause compared to the dynamic tropopause. This may be the result of
the thermal tropopause being located at a higher altitude than the dynamical tropopause (Petzold et al., 2020), for which drier
conditions may be observed with this definition. This is also inline with reports by Petzold et al. (2020), who found a lower
ISSR fraction in the lower stratosphere when using the thermal tropopause definition compared to the dynamic tropopause,
indicating more moist conditions with the latter definition. We also find that the thermal tropopause is drier in the upper
troposphere in comparison to the dynamic tropopause.

This analysis shows the results are impacted by the choice of the tropopause definition. However, in this study, we are inter-
ested in the differences that exist between IAGOS and ERA5. Hence, due to the sharper gradients for the thermal tropopause,
we will use this definition for the study.



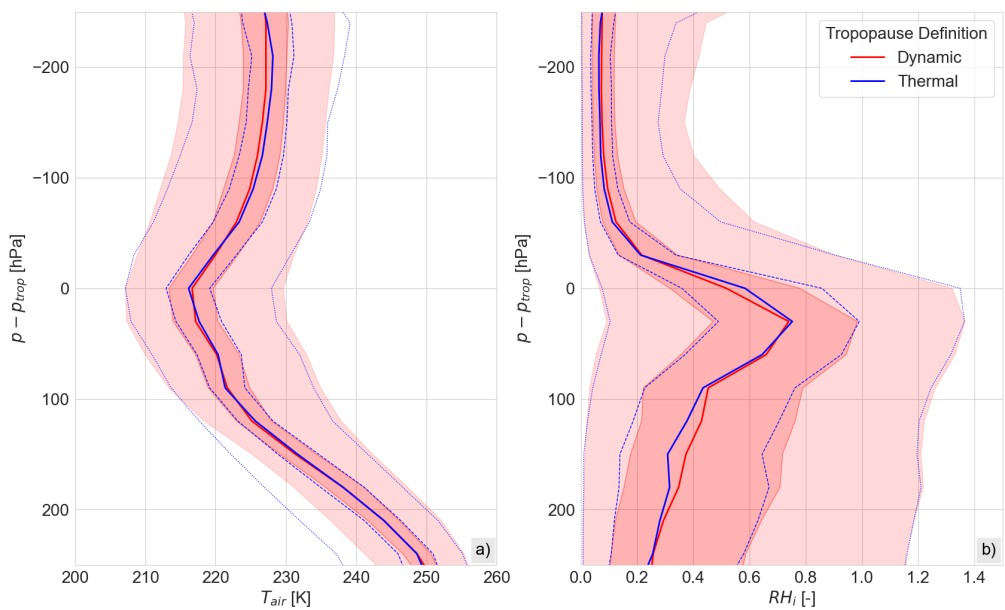

**Figure A1.** Vertical distribution of **a)** temperature and **b)** RHi for dynamic and thermal tropopause definitions using all considered IAGOS measurements in this study. Solid line indicates the mean, dashed (blue) line and dark (red) shade indicate the 25 and 75% percentiles, and dotted (blue) line and light (red) shade indicate the 1 and 99% percentiles.

*Author contributions.* KGH and FY designed the study. KGH performed the data analysis and prepared the manuscript under the supervision of FY. FC applied the North Atlantic weather pattern classification. FC, VM, FY and KGH reviewed and edited the manuscript.

*Competing interests.* The contact author has declared that none of the authors has any competing interests.

*Acknowledgements.* We acknowledge the strong support of the European Commission, Airbus, national agencies in Germany (BMBF), France (MESR), and the UK (NERC), and the IAGOS member institutions (http://www.iagos.org/partners), and the airlines (Lufthansa, Air France, Austrian, China Airlines, Hawaiian Airlines, Air Canada, Iberia, Eurowings Discover, Cathay Pacific, Air Namibia, Sabena) that have carried the MOZAIC or IAGOS measurement equipment free of charge and performed maintenance since 1994. The data are available at http://www.iagos.fr thanks to additional support from AERIS. IAGOS has been funded by the European Union projects IAGOS–DS and
IAGOS–ERI, INSU-CNRS (France), Météo-France, Université Paul Sabatier (Toulouse, France), and Forschungszentrum Jülich (FZJ, Jülich, Germany).



*Financial support.* This research has received funding from the Horizon Europe Research and Innovation Actions programme under Grant Agreement No 101056885 and from Horizon Europe Research and Innovation Actions programme under Grant Agreement No 101167020.



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
