# Peer review of "Variability of ice supersaturated regions at flight altitudes: evaluation of ERA5 reanalysis using IAGOS in situ measurements"

_EGUsphere, 2025_

## Referee Comment (RC1)

**Review of „Variability of ice supersaturated regions at flight altitudes: evaluation of ERA5 reanalysis using IAGOS in situ measurements by Hildebrandt et al., 2025**

The manuscript tackles an important topic, namely the representation of ice supersaturated regions (ISSRs) in ERA5 and their comparison with in situ observations from IAGOS. The study is not restricted to the North Atlantic but also includes several other regions, including the tropics. The paper is generally well written and technically sound.
That said, the degree of novelty is limited. Key results such as the ERA5 dry bias, the underestimation of ISSR occurrence, and the generally low ETS have already been documented in recent studies. While the authors expand the analysis in both spatial and temporal scope—which is certainly valuable—the added contribution beyond confirming previous findings is not fully evident.

Therefore, I recommend **major revisions** before the manuscript can be accepted.

**Major Comments**

This paper presents in detail the results of the comparison between ERA5 and IAGOS with regard to relative humidity over ice. Most of the results confirm earlier studies and do not provide any truly new insights. Additionally, the paper is very long.

It should be noted, however, that the tropics are also coming more into focus here. Perhaps this would be a way to make the manuscript more exclusive. I think there are many interesting findings in it, but more attention should be paid to the influence of aircraft routes on the results.
For example, while in the Southern Trans-Pacific region most of the measurement points are north of the equator and only at the edges of the area, the South Atlantic and Africa cover a very good area in a north-south direction up to 30°S. How does this influence the results?
This also raises the question of whether the division of the seasons into DJF, MAM, JJA and SON makes sense here. Perhaps the tropics should be evaluated using different methods than the extra tropics?

Many methods are taken from previous work, such as the distinction between cloud free and cloudy conditions. It would be interesting to see how the results change when these definitions change. Or are there perhaps even better methods for doing this?

Another example that comes to mind is the use of ETS. Although ETS is a frequently used measure in meteorology and is also used in the studies cited, I wonder how meaningful the results are or what added value they provide when we already know that ERA5 has problems with ice supersaturation.
How do the results change if ISSRs in ERA are defined as, for example, RHi=90%? Similarly, in the later analysis in Sec 3.5: how sensitive are the results to the thresholds used to distinguish between cloudy and clear skies?
Could there be another measure that does not penalize spatial or temporal shifts as heavily as ETS?

The consideration of weather patterns over the North Atlantic does not fit well into the manuscript. Why are other patterns over different regions not also taken into account? First, it is pointed out that many other regions are also being studied, only to then undertake a very specific study of the North Atlantic. This part is also rather shallow.

Some of the figures are difficult to read, and fine details are sometimes hardly visible. The axis labels are also incorrect in some cases (saturation ration or RHi?). More emphasis should be placed on consistency (always the same order in the legend, each subplot with vertical axis labels or only the first column?).

**Specific comments**

Line 5: „data set" missing after *reanalysis*?

Line 16: Use another word than *important*. Maybe „significant"?

Line 37: I think ERA-Interim was compared to IAGOS

Line 100 & 108: Formatting of reference is broken

Line 145: how were RHi and RHI calculated in detail? This is an important information

Line 150f: Why is the stronger gradient an argument for the use of the thermal tropopause? This is not clear to me. Did you consider the cold point tropopause (CPT) for the tropics?

Figure 2: X-axis bottom row: What kind of frequency? Relative to what?

Line 168 & 180: What does „no measurement" mean? Data point of a flight without BCP? Please clarify.

Figure 3: See Fig. 2: What kind of frequency relative to what?

Line 175: Why was the approach chosen from Wolf et al. (2025) using the cloud cover instead of the IWC used in Wang et al. (2025)? This definition is crucial to the results later on. How sensitive are the results to these values?

Sec. 2.5: What was the influence of the COVID-19 pandemic on the statistics?
During the pandemic, the number of commercial flights decreased substantially. Did this reduction in flight activity affected the amount of IAGOS data collected during that period. How does this reduced sampling influence the representativeness of the results, in particular for the distribution across different weather patterns? So e.g. Winter patterns 1 and 4 show high values in Winter 2020/2021.
How many measurements fall into each weather pattern?

Line 207: „*maybe due to sensor error*": be more specific.

Line 219: what is meant by „*better approximation of temperature*"?

Figure 7: shading of 95% confidence interval not visible. Also, the order of the seasons is different in the legend from figure to figure (JJA, SON, DJF, MAM, in Fig 7 but DJF, MAM… in Fig. 12)

Line 225f and Figure. 7: Why are the minima of temperatures for the tropic cases below the thermal tropopause and not as distinct as in the extra tropics? Also not only JJA in South Asia is below the thermal tropopause. The minimum of DJF is even lower (approximately 900 hPa, Fig. 7e). You only discuss it for JJA in South Asia.

Figure 8: X-axis. Either Saturation ratio without units or RHi with [%]. You show values for saturation ratio not RHi.

Line 245: „distribution" missing after *vertical*?

Line 257: I don't understand the sentence starting with „The lower values of RHi…."?

Figure 9: X-axis: Same as Fig. 8. You show values for saturation ratio, not RHi.

Line 261. Be more precise. What do you mean with the *highest RHi*? Peak value? Or throughout the profile?

Line 272: I do not agree. I think a seasonal cycle is also clearly visible for Fig. 9 e)

Line 273: Is the argument for the high RHi values in South Asia in JJA due to deep convection not also valid for other tropical regions? Why is it so different for South Asia? Is it an observation of the meteorological characteristics of the region or rather due to the course of the flight routes in this area?

Line 275f: where can I see this?

Line 302. The beginning of this sentence sounds odd. Why should it be dry in the first place?

FIgure 10: saturation ratio or RHi

Figure 11: Subfigure e): Are there no cloudy or indeterminate measurements from IAGOS? I can't see solid lines for this cases.

Line 340/341: *ERA5 may predict less … than ERA5…*

Figure 12: subplot f): There are no labels for height levels above the tropopause. In g) there is only „-50".

Line 386: „*outliers*": What might be the reason for these outliers? Physical or sampling? I think this should be more specific.

General: The figures are sometimes hard to read with a lot of different lines. Also please check if your are plotting saturation ratio or RHi!

Line448/449: The reader is left alone with the question raised. What would the authors' suggestions be?

Line 457: I'm not sure I can agree with the authors' conclusion here. In some cases, the weather patterns cover different altitudes but are still all within a comparable range.

Line 476f: It should be clarified whether the lower performance of ERA-Interim implies that the seasonality found by Irvine et al. (2012) is no longer valid, or whether the differences mainly reflect dataset characteristics and methodological choices. The current phrasing ("but") may unintentionally suggest that the earlier results are invalid.

Line 550: You recommend a standard method but do not specify which one.

Code availability: I strongly recommend making the code publicly available via platforms such as zenodo.

Line 588: Which time frame?

Line 605: Why is a sharper gradient the basis for selection?

Figure A1: Saturation ratio or RHi?

Figure S1:  The important Information is very small in this figure, grey shading is not visible.

Figure S6: What is the x-Axis? Is ΔRHi in units of RHi? Or is it a relative difference between RHi_iagos and RHi_era?

---

## Author Comment (AC1)

**Reply to Reviewer #1**

We would like to thank the reviewer for their constructive comments, which have helped improve the paper significantly. Based on the review, we have shortened the paper to focus more on the new results, particularly those for the tropics. While revising the paper, we found an error in the plot of the vertical distribution of the mean temperature (Fig. 7 in the preprint), where the mean temperature was plotted with respect to the dynamic tropopause instead of the thermal tropopause. This only affected the interpretation of the results in Sect. 3.1, which has been corrected. However, it does not change the overall conclusion that ERA5 shows a cold bias in temperature when compared to IAGOS. In the following, we reply to the individual comments from the reviewer. We use colour to organize this as follows:

- The questions and comments from the reviewers are marked in blue.
- The replies from the authors are written in black.
- Any changes to the manuscript are written in green.

Major Comments
**Reviewer:** This paper presents in detail the results of the comparison between ERA5 and IAGOS with regard to relative humidity over ice. Most of the results confirm earlier studies and do not provide any truly new insights. Additionally, the paper is very long.
**Authors:** Thank you for the feedback. The comparison between ERA5 and IAGOS indeed focuses a lot on confirming earlier studies, which is a way for us to evaluate our analysis approach. Nevertheless, we agree that the paper is perhaps too long. Following the suggestions, we have shortened the paper by focusing more on novel results, i.e. those from the tropics and removing comparisons to literature in the results section. Comparisons to literature are now mainly found in a newly added discussion section (Sect. 4). Furthermore, we have removed Sect. 2.5 on the North Atlantic weather pattern methodology and Sect. 3.6 on the analyses of the North Atlantic weather patterns based on the other feedback received. These sections are now found in Sect. S7 in the supplement. We suggest it as future research in the discussion section. We have also removed Fig. 5 from Sect. 2.4, which showed the global mean annual cirrus occurrence, and Fig. 8 from Sect. 3.2, that showed a 2D histogram of the ERA5 RHi as a function of IAGOS RHi; we found that their key messages were mainly confirming previous results.

**Reviewer:** It should be noted, however, that the tropics are also coming more into focus here. Perhaps this would be a way to make the manuscript more exclusive. I think there are many interesting findings in it, but more attention should be paid to the influence of aircraft routes on the results. For example, while in the Southern Trans-Pacific region most of the measurement points are north of the equator and only at the edges of the area, the South Atlantic and Africa cover a very good area in a north-south direction up to 30°S. How does this influence the results? This also raises the question of whether the division of the seasons into DJF, MAM, JJA and SON makes sense here. Perhaps the tropics should be evaluated using different methods than the extra tropics?
**Authors:** Thank you for your insightful suggestions. We agree that the analysis in the tropics adds new insights next to the available literature. More interpretations on the tropics analysis have been included in the revision.

We have done an extensive literature review on whether the division of the seasons into DJF, MAM, JJA and SON is suitable in the tropics, as we are aware that the tropics are divided into wet and dry

seasons. The distinction between wet and dry season is influenced by many factors. One factor is the inter-tropical convergence zone (ITCZ), which causes changes in wet and dry conditions across the equator depending on its position (National Oceanic and Atmospheric Administration, 2023a). The position of the ITCZ is seasonally dependent as it follows the variation in incoming solar radiation (National Oceanic and Atmospheric Administration, 2023a).

We researched if there were any methodologies for classifying the wet and dry seasons in the tropics. On possible option was the Köppler-Geiger climate classification, which globally describes the precipitation and temperature patterns (Murray et al., 2007; National Oceanic and Atmospheric Administration, 2023b). The precipitation patterns are categorised as wet year-round, dry summer season, dry winter season and monsoon (National Oceanic and Atmospheric Administration, 2023b). This can help determine which seasons are wet and which are dry in the tropical regions we have defined. However, we find that it is not straightforward to use the Köppler-Geiger climate classification as our regions might encompass both a dry summer and a dry winter region, or a region which is wet all year round. Thus, while the classification of wet and dry seasons could be useful, it is difficult when using larger general regions, which encompass varying climates.

Based on the ITCZ and the aircraft routes in the tropics, we also analysed the impact of considering different latitudinal bands in the defined tropical regions. We considered the bands 0 to $30^O$N and 0 to $30^O$S as the ICTZ causes changes in conditions across the equator, while keeping the same longitudinal bands for the defined tropical regions. By considering separately the two hemispheres, different seasonal behaviours of RHi emerge. For example, in South Asia, considering the latitudinal band 0 to $30^O$N result in the highest RHi values throughout the vertical profile occurring during JJA, while this peak occurs in DJF when considering the 0 to $30^O$S band. The change in seasonality of RHi considering separately the Northern and Southern Hemisphere could be motivated by the ICTZ.

Hence, our recommendation is to still consider the division of the seasons into DJF, MAM, JJA and SON in the tropics, but consider Northern and Southern hemisphere tropics separately. However, due to low sampling in the Southern Hemisphere tropics, we choose to only focus on the Northern Hemisphere. We have updated all figures accordingly and added Appendix A as a supporting argument.

**Reviewer:** Many methods are taken from previous work, such as the distinction between cloud free and cloudy conditions. It would be interesting to see how the results change when these definitions change. Or are there perhaps even better methods for doing this?

**Authors:** Thank you for this comment. As a sensitivity study, we have compared the results using different definitions to classify in-cloud and clear-sky conditions, based on the cloud cover (CC) as in Wolf et al. (2025), or based on the cloud ice water content (CIWC), as in Wang et al. (2025) and, more recently, in Petzold et al. (2025). Indeed, we find that the results change when these definitions change. For example, if we use the CIWC based on Petzold et al. (2025) to classify both IAGOS and ERA5 measurements, we find that ERA5 shows a much better ability of predicting ice supersaturated regions under cloudy conditions compared to the original definitions we used in our paper. However, the CIWC definition also shows a decrease in the performance of ERA5 to predict ISSRs in clear-sky conditions compared to the definitions in the paper. This is because the CIWC has a much lower probability of ice supersaturation conditions under clear skies as the thresholds defined remove the 'shoulder' of the RHi PDF that was seen under clear skies with the cloud cover and ice crystal number concentration. Hence, it seems that either the definition shows

that ERA5 has good ability to predict ISSRs under clear-sky conditions or under cloudy conditions, but not both. Our findings on the effect of the definitions on the results can be found in the supplement (Sect. S1).

**Reviewer:** Another example that comes to mind is the use of ETS. Although ETS is a frequently used measure in meteorology and is also used in the studies cited, I wonder how meaningful the results are or what added value they provide when we already know that ERA5 has problems with ice supersaturation. How do the results change if ISSRs in ERA are defined as, for example, RHi=90%? Similarly, in the later analysis in Sec 3.5: how sensitive are the results to the thresholds used to distinguish between cloudy and clear skies? Could there be another measure that does not penalize spatial or temporal shifts as heavily as ETS?

**Authors:** Yes, it is indeed already known that ERA5 has problems with capturing ice supersaturation. To our best knowledge, these issues have mainly been documented in the Northern Hemisphere, i.e. the North Atlantic (Teoh et al., 2022),  Europe and Africa (Wang et al., 2025). In these geographical regions, calculating the ETS may not result in new information, since similar analyses have been previously presented e.g. Gierens et al. (2020), Teoh et al. (2022) and Wang et al. (2025). Whereases, we provide new information on how the ETS changes per season, which was only previously done in the study by Gierens et al. (2020) based on a dataset using 4 months in 2014, whereas we use more than 10 years of data. If we have overlooked any studies in other geographical regions, such as South Asia, we are open to adapt.

On the other hand, we agree that it is interesting to examine how the ETS changes if the ISSRs in ERA5 are defined with different RHi thresholds. Hence, in Sect. 3.4, we have changed the analysis on the vertical distribution of the ETS per season and geographical region to how the ETS changes with different RHi thresholds. We considered a total of six different thresholds,  ranging from 90% to 102%, as we found a moist bias in South Asia. We have also done a similar analysis for the cloudy and clear-sky conditions.

In terms of another measure, neighbourhood methods could perhaps account for spatial and/or temporal shifts by evaluating ERA5 within a spatial and or temporal window surrounding each IAGOS measurement, according to Jolliffe & Stephenson (2012). While we cannot consider a complete window as our ERA5 data has been collocated to IAGOS flight tracks, we could consider neighbouring measurement points along a flight path, i.e. a neighbourhood equitable threat score (Schwartz, 2017). The neighbourhood ETS for season DJF and JJA, considering neighbouring measurements points along a flight path, are shown below in Figure 1 and Figure 2.  We considered various ERA5 RHi thresholds in ERA5 (same as Figure 10 and Figure 11 in the revised paper). While we do see some minor improvements in the ETS, values are still in the same range and indicate the same correlations as previously observed. Hence, due to the nature of our dataset, it is difficult to use another measure, where we do not penalise the spatial and temporal shifts.

[Figure]

Figure 1: Vertical distribution of ice supersaturated region equitable threat score per geographical region for different ERA5 RHi thresholds in DJF.

[Figure]

Figure 2: Vertical distribution of ice supersaturated region equitable threat score per geographical region for different ERA5 RHi thresholds in JJA.

**Reviewer:** The consideration of weather patterns over the North Atlantic does not fit well into the manuscript. Why are other patterns over different regions not also taken into account? First, it is pointed out that many other regions are also being studied, only to then undertake a very specific study of the North Atlantic. This part is also rather shallow.

**Authors:** Thank you for the feedback and for raising this question. The choice of only considering the weather patterns over the North Atlantic is because it is the only region in our study which is defined by typical large-scale weather patterns at flight altitudes; we are currently not aware of such definitions at flight altitudes for the other regions. We agree that this section requires more analysis and can be an independent study on its own. Therefore, we have removed it as part of the results and incorporated it into a discussion section to suggest it as future research. This also means that we have removed Sect. 2.5 and Sect. 3.6, on the methodology and results for the North Atlantic weather pattern, which is now in the supplement (Sect. S7) to support the discussion.

**Reviewer:** Some of the figures are difficult to read, and fine details are sometimes hardly visible. The axis labels are also incorrect in some cases (saturation ration or RHi?). More emphasis should be placed on consistency (always the same order in the legend, each subplot with vertical axis labels or only the first column?).

**Authors:** We appreciate the feedback, and we have taken it into account for all figures. All axis labels on all figures related to the RHi have been fixed to be shown in percentages. This aspect is now consistent in the text as well. We have also ensured to use the same order in the legend for both the season and cloudy condition labels. Further, we have also streamlined that in all subplots, only the first column has vertical axis labels.

Specific comments

**Reviewer:** Line 5: „data set" missing after reanalysis?

**Authors:** Thank you for spotting this; this change has been incorporated.

**Reviewer:** Line 16: Use another word than important. Maybe „significant"?

**Authors:** Good suggestion; We have changed "important" to "significant".

**Reviewer:** Line 37: I think ERA-Interim was compared to IAGOS

**Authors:** Thank you for this comment. Indeed, the paper mentions comparing ERA-interim to IAGOS. However, the paper used IAGOS data from January 2000 to December 2009, which is technically MOZAIC (August 1994 to December 2014). We attempt to make the distinction between IAGOS and MOZAIC based on the date alone. We have edited the sentence to better incorporate this. This is the new sentence:

"This leads to biases in the prediction of relative humidity in weather models. Reutter et al. (2020) showed that ERA-Interim, the predecessor of ERA5, underestimated RHi when greater than 100%, leading to an underestimation in the occurrence of ISS when compared to MOZAIC (Measurement of OZONE and Water Vapour on Airbus in-service Aircraft), which is currently part of IAGOS."

**Reviewer:** Line 100 & 108: Formatting of reference is broken

**Authors:** Thank you for pointing this out. The reference has been fixed.

**Reviewer:** Line 145: how were RHi and RHl calculated in detail? This is an important information

**Authors:** We agree that this is important information. Initially, we used the Sonntag (1994) ice saturation pressure equation, which is valid for the temperature range considered in our analysis of ERA5 and IAGOS and has been used in other contrail studies (Schumann, 1996, Schumann, 2012, Roosenbrand et al., 2023).

We realise that this is not the same formulation used in the ECMWF, which uses the AERKI formula of Alduchov and Eskridge (1996). These two formulations do result in differences in RHi. For example, the ECWMF formulation results in slightly higher values of RHi, which can lead to more points above the ice supersaturation threshold. This was also reflected in a slightly higher ISSR fraction with the ECMWF formulation compared to Sonntag, but the difference was small. If we use the Sonntag (1994) formulation, the ISSR fraction of our entire ERA5 dataset is 7.9%. Using the ECMWF formulation increases the ISSR fraction to 8.0% for our entire ERA5 dataset. Hence, using different ice saturation pressure equations result in a minor difference in RHi. Nevertheless, we

have decided to update our formulation to calculate the RHi using the AERKI formula of Alduchov and Eskridge (1996) and the details are now included in Sect. 2.2.

**Reviewer:** Line 150f: Why is the stronger gradient an argument for the use of the thermal tropopause? This is not clear to me. Did you consider the cold point tropopause (CPT) for the tropics?

**Authors:** Thank you for the comment. In the revised paper, we have removed the sentence "Hence, due to the sharper gradients for the thermal tropopause, we will use this definition for the study". The paragraph now reads as follows:

"This analysis shows that the results are impacted by the choice of the tropopause definition. However, in this study, we are interested in the differences between IAGOS and ERA5, and are interested in the separation of dry and moist conditions. Although there are no significant differences in the results when using the thermal and dynamic tropopause definitions, there is a tendency for the dynamic tropopause definition to result in higher values of RHi in the lower stratosphere compared to the thermal tropopause definition. Therefore, we decided to use the thermal tropopause to ensure better separation of moist and dry conditions."

Indeed, the cold point tropopause might be relevant for the tropics. However, the tropopause height in the tropics (16 km – 17 km; Hoffmann & Spang, 2022) is generally above the typical cruising altitude of passenger aircraft; using the data on IAGOS aircraft collected for this paper, we found that IAGOS aircraft most frequently visited altitudes between 8 km and 13 km. This is also reflected in Figure 2 in the paper, which shows the vertical distribution of the number of IAGOS measurements. Comparison of the temperature in the tropics using the dynamic and thermal tropopause reveals many similarities. If we used the cold point tropopause, the tropopause would generally be at or above the WMO 1$^{st}$ thermal tropopause (Mehta et al., 2011), which is the definition used in our paper. It should also be noted that the maximum altitude of the IAGOS measurements in the tropics is around 12500 meters, which is much lower than the reported cold point tropopause (CPT) height by Metha et al. (2011), where the CPT height in the tropics was reported to be between approximately 17 and 18.5 km. Therefore, changing the tropopause definition will most likely not have a large influence on our results as most measurements are already in the upper troposphere.

**Reviewer:** Figure 2: X-axis bottom row: What kind of frequency? Relative to what?

**Authors:** Thank you for the question. By frequency in Figure 2, we mean to define the fraction of the dataset used within the study, i.e the filtered and sampled dataset. The x-axis of Figure 2 has been updated to 'Fraction of dataset' to better define this.

**Reviewer:** Line 168 & 180: What does „no measurement" mean? Data point of a flight without BCP? Please clarify.

**Authors:** Yes, "no measurement" means that there are no measurements available for number of ice particles available from IAGOS. This has been clarified in captions of Figure 3 and Figure 4.

**Reviewer:** Figure 3: See Fig. 2: What kind of frequency relative to what?

**Authors**: In Figure 3, we mean to describe the frequency of occurrence of different conditions (cloudy, clear-sky and indeterminate) as the fraction of the level for each respective region where such conditions occur. We have updated Figure 3 to better define the frequency. Subsequently, we have made the same changes to Figure 4. The x-axis of Figure 3 and Figure 4 is now labelled as 'Fraction of samples per level'.

**Reviewer:** Line 175: Why was the approach chosen from Wolf et al. (2025) using the cloud cover instead of the IWC used in Wang et al. (2025)? This definition is crucial to the results later on. How sensitive are the results to these values?

**Authors:** Thank you for this question. We have discussed this sensitivity addressing the major comment "Many methods are taken from previous work, such as the distinction between cloud free and cloudy conditions. It would be interesting to see how the results change when these definitions change. Or are there perhaps even better methods for doing this?"

Hence, we copy here the relevant text:

As a sensitivity study, we have compared the results using different definitions to classify in-cloud and clear-sky conditions, based on the cloud cover (CC) as in Wolf et al. (2025), or based on the cloud ice water content (CIWC), as in Wang et al. (2025) and, more recently, in Petzold et al. (2025). Indeed, we find that the results change when these definitions change. For example, if we use the CIWC based on Petzold et al. (2025) to classify both IAGOS and ERA5 measurements, we find that ERA5 shows a much better ability of predicting ice supersaturated regions under cloudy conditions compared to the original definitions we used in our paper. However, the CIWC definition also shows a decrease in the performance of ERA5 to predict ISSRs in clear-sky conditions compared to the definitions in the paper. This is because the CIWC has a much lower probability of ice supersaturation conditions under clear skies as the thresholds defined remove the 'shoulder' of the RHi PDF that was seen under clear skies with the cloud cover and ice crystal number concentration. Hence, it seems that either the definition shows that ERA5 has good ability to predict ISSRs under clear-sky conditions or under cloudy conditions, but not both. Our findings on the effect of the definitions on the results can be found in the supplement (Sect. S1).

**Reviewer:** Sec. 2.5: What was the influence of the COVID-19 pandemic on the statistics? During the pandemic, the number of commercial flights decreased substantially. Did this reduction in flight activity affected the amount of IAGOS data collected during that period. How does this reduced sampling influence the representativeness of the results, in particular for the distribution across different weather patterns? So e.g. Winter patterns 1 and 4 show high values in Winter 2020/2021. How many measurements fall into each weather pattern?

**Authors:** Thank you for this question. We investigated the influence of the COVID-19 pandemic on the statistics. As seen in Fig. 3 below of the number of measurements per year when considering our entire IAGOS dataset, the COVID-19 pandemic clearly caused a decrease in the number of measurements in 2020. However, already in 2021 it started to readjust to pre-COVID-19 levels.

[Figure]

*Figure 3: Number of IAGOS measurements per year*

It should be noted that Figure 6 in the preprint submission, which showed the frequency of weather patterns per year using ERA5, is not a representation of the number of IAGOS measurements, but it reflects the weather over the North Atlantic for the years considered in our study, i.e. it considers all days in winter and summer from year 2011 to 2022, regardless of whether an IAGOS measurement was taken on that given day.

Nevertheless, as discussed above, the focus of this study is not on the yearly variability; thus Figure 6 does not align well with the rest of the contents of the paper. Therefore, we have removed Figure 6 from the paper completely, but it has been replaced by Fig. 4 below showing the vertical distribution of the number of measurements within each weather pattern. Note, this new figure is now found in the supplement (Sect. S7) as this is where we have moved Section 2.5, as previously noted.

[Figure]

*Figure 4: Number of IAGOS measurements for each North Atlantic weather pattern*

**Reviewer:** Line 207: „maybe due to sensor error": be more specific.
**Authors:** We have modified this sentence to be more specific, while we have also shortened the overall text in this section. Hence, it now reads as follows:
"In the extratropics, we find that ERA5 has a cold bias of 0.5 K on average in the upper troposphere, but it cannot be ruled out that this is not due to sensor error as the mean difference is within accuracy range of the IAGOS temperature sensor."

**Reviewer:** Line 219: what is meant by „better approximation of temperature"?
**Authors:** We agree that this sentence is not entirely clear. We meant to say the following:
"Meanwhile, in the tropical regions, there is a smaller temperature difference between IAGOS and ERA5 below the tropopause, showing that ERA5 better approximates temperature in the tropical upper troposphere." However, to shorten the paper, we have removed this sentence entirely.

**Reviewer:** Figure 7: shading of 95% confidence interval not visible. Also, the order of the seasons is different in the legend from figure to figure (JJA, SON, DJF, MAM, in Fig 7 but DJF, MAM… in Fig. 12)
**Authors:** Thank you for pointing this out. Indeed, the shading is not visible because the 95% confidence interval is already small.
We appreciate the comment on the order of seasons in the legend. This has now been changed to fit the same order as presented in Fig. 12. Furthermore, we have ensured that all figures which show the seasons have the same order of the seasons in the legend (Figure 7, Figure 9 and Figure 12, considering the labelling in the preprint).

**Reviewer:** Line 225f and Figure. 7: Why are the minima of temperatures for the tropic cases below the thermal tropopause and not as distinct as in the extra tropics? Also not only JJA in South Asia is below the thermal tropopause. The minimum of DJF is even lower (approximately 900 hPa, Fig. 7e). You only discuss it for JJA in South Asia.

**Authors:** Thank you for this question. While reviewing the paper, we found an error in the plot of the vertical distribution of the mean temperature (Fig. 7 in the preprint), where the mean temperature was plotted with respect to the dynamic tropopause instead of the thermal tropopause. This is why the minima of temperatures for the tropics was below the tropopause. This error was isolated to Fig.7 in the preprint and thus only affected the interpretation of the results in Sect. 3.1. The plot has now been updated to correctly use the thermal tropopause, for which the behaviour of the temperature profile now appears to better match with this tropopause definition. The corresponding text has also been corrected. However, it does not change the overall conclusion that ERA5 shows a cold bias in temperature when compared to IAGOS. For example, we still find ERA5 to have a cold bias of 0.5 K on average in the extratropical upper troposphere.

**Reviewer:** Figure 8: X-axis. Either Saturation ratio without units or RHi with [%]. You show values for saturation ratio not RHi.
**Authors:** Figure 8 has been adapted to show RHi with [%].

**Reviewer:** Line 245: „distribution" missing after vertical?
**Authors:** This has been incorporated.

**Reviewer:** Line 257: I don't understand the sentence starting with „The lower values of RHi...."?
**Authors:** We revised the sentence as follows for better readability:
"This indicates that under conditions where RHi is lower, there is better agreement between IAGOS and ERA5..."

**Reviewer:** Figure 9: X-axis: Same as Fig. 8. You show values for saturation ratio, not RHi.
**Authors:** Figure 9 has been adapted to show RHi with [%].

**Reviewer:** Line 261. Be more precise. What do you mean with the highest RHi? Peak value? Or throughout the profile?
**Authors:** Thank you for raising this question. We have reworded it, while also updating Fig.7 to Fig. 5 as some figures have been removed. The sentence is now:
"In North America, North Atlantic, and Europe, the highest RHi values throughout the vertical profile occur during seasons DJF and MAM, which are also the seasons that exhibit the lowest mean temperatures, as seen in Fig 5."

**Reviewer:** Line 272: I do not agree. I think a seasonal cycle is also clearly visible for Fig. 9 e)
**Authors:** We agree that some seasonal cycle is visible for Fig 9 e) and Fig 9 h), especially after limiting the analysis to only consider the tropics in the Northern Hemisphere. We mention the seasonal cycle in our analysis now:
"In the tropics, we observe some seasonal behaviour in the Southern Trans-Pacific and in South Asia. South Asia has a larger mean RHi in season JJA compared to the other seasons in this region...

In the Southern Trans-Pacific, season DJF shows an overall lower mean RHi compared to other seasons..."

**Reviewer:** Line 273: Is the argument for the high RHi values in South Asia in JJA due to deep convection not also valid for other tropical regions? Why is it so different for South Asia? Is it an observation of the meteorological characteristics of the region or rather due to the course of the flight routes in this area?

**Authors:** Thank you for asking this question. Indeed, deep convection can also happen in the other tropical regions. However, we see a lot more measurements classified as cloudy in South Asia compared to in the other tropical regions. Hence, it is most likely due to the course of the flight routes in this area. We have included this sentence in the paragraph:

"Although deep convection can also happen in other tropical regions, we do not observe this due to the low sampling of in-cloud measurements with IAGOS in these regions. "

**Reviewer:** Line 275f: where can I see this?
**Authors:** Thank you for your interest in this information. Originally, it was not included due to the length of the paper. However, we have decided to add it in the supplement (Sect. S4).

**Reviewer:** Line 302. The beginning of this sentence sounds odd. Why should it be dry in the first place?
**Authors:** Yes, we agree with this, and it was also pointed out by reviewer #2. This sentence has been removed in the restructuring of this section. However, we still include a suggestion from reviewer #2. The part related to this original line now reads as follows:

"The UT has more moisture compared to the TROP and LS, allowing for higher probabilities of ISS occurrence. However, the TROP also shows high ISS occurrence, both in the tropics and extratropics. This is in line with Reutter et al. (2020), where it was found that there was a significant amount of in-situ and ERA-interim data that exceeded the RHi = 100% threshold near the tropopause in the North Atlantic flight corridor."

**Reviewer:** FIgure 10: saturation ratio or RHi
**Authors:** Figure 10 has been adapted to show RHi with [%].

**Reviewer:** Figure 11: Subfigure e): Are there no cloudy or indeterminate measurements from IAGOS? I can't see solid lines for this cases.
**Authors:** That is correct. Looking at Figure 3: subfigure e), it can also be seen that the majority of IAGOS datapoints in the Southern Trans-Pacific region at the tropopause are either in clear-sky conditions or do not have any measurements for the number of ice particles. We have also adapted Figure 11 to show RHi with [%].

**Reviewer:** Line 340/341: ERA5 may predict less ... than ERA5...
**Authors:** Thank you for pointing this out, this sentence has been corrected.

**Reviewer:** Figure 12: subplot f): There are no labels for height levels above the tropopause. In g) there is only„-50".
**Authors:** Figure 12 has been adapted such that the y-axis is shared between all subplots and no labels for height levels are missing above the tropopause.

**Reviewer:** Line 386: „outliers": What might be the reason for these outliers? Physical or sampling? I think this should be more specific.
**Authors:** Thank you for raising this question. We have analysed the outliers in more detail and we have adapted this part to be more specific as to the cause of the outliers. This is what we have changed/added:

"There are a few instances of high ETS scores: North America in season JJA at 180 hPa below the tropopause and North Atlantic in season DJF at 150 hPa below the tropopause. For North America in season JJA at 180 - $p_{thm}$ hPa, there is a low number of ISSRs in both IAGOS and ERA5 (see Fig. 9)

where one measurement point is classified as TP, one point as FN and the rest as TN. Since the ISSR events are rare, resulting in a perfect hit rate, accompanied with most points classified as TN, the ETS can inflate. We see a similar low ISSR fraction for North America in season JJA at 150 - $p_{thm}$ hPa, although the ETS is not as high. This can be attributed to more points being categorised as FN and FP, lowering the ETS. In the North Atlantic, in season DJF at 150 hPa below the tropopause, we also find a high ETS. In the instances where an ISSR is observed here, most of these events are classified as TP, with no misses (FN = 0), and there were few points classified as FP compared to TP. At the same time, there are also many points classified as TN. The combination of this leads to a high ETS."

**Reviewer:** General: The figures are sometimes hard to read with a lot of different lines. Also please check if your are plotting saturation ratio or RHi!
**Authors:** We have adapted all plots with RHi to show RHi with [%].

**Reviewer:** Line448/449: The reader is left alone with the question raised. What would the authors' suggestions be?
**Authors**: We agree that the original sentence ends on a cliffhanger. We have now revised it to provide a clear explanation of our recommendation, which is also based on our sensitivity study: "As discussed in Sect. S1 in the supplement, the cloud ice water content definition is limited in identifying clear-sky ice supersaturated conditions. In contrast, the ice crystal number concentration in IAGOS and the cloud cover definition in ERA5 are more suitable for analysing ice supersaturation in clear-sky and cloudy conditions. Hence, we recommend using the latter definition."

**Reviewer:** Line 457: I'm not sure I can agree with the authors' conclusion here. In some cases, the weather patterns cover different altitudes but are still all within a comparable range.
**Authors:** Thank you for this comment. We agree that there is not a large difference between the ETS for the different winter weather patterns in the North Atlantic. In the end, with moving the North Atlantic weather pattern to the discussion as it does require further research, this line/sentence has been removed.

**Reviewer:** Line 476f: It should be clarified whether the lower performance of ERA-Interim implies that the seasonality found by Irvine et al. (2012) is no longer valid, or whether the differences mainly reflect dataset characteristics and methodological choices. The current phrasing ("but") may unintentionally suggest that the earlier results are invalid.
**Authors:** We agree that using 'but' may unintentionally suggest that earlier results are invalid, which is not what we intended to do. Due to the restructuring of the paper, this sentence has been removed in its entirety.

**Reviewer**: Line 550: You recommend a standard method but do not specify which one.
**Authors:** We have included our recommendation:
"We recommend to use the ice crystal number concentration from IAGOS and the cloud cover from ERA5 for comparing ISSRs under cloudy and clear-sky conditions between the two datasets due to the limitations of the cloud ice water content under clear-sky conditions."

**Reviewer**: Code availability: I strongly recommend making the code publicly available via platforms such as zenodo.

**Authors:** The code will be made available at 10.4121/915dc859-c397-4294-8e85-83451ebaa881 after publication.

**Reviewer:** Line 588: Which time frame?
**Authors:** We considered the entire timeframe of our study. We have adapted the text to include this, while Figure A1 is now Figure B1 due to the addition of another appendix. It now reads as follows:
"Figure B1 shows the vertical distribution of temperature and relative humidity over ice for the thermal and dynamic tropopause definitions from July 2011 to December 2022."

**Reviewer:** Line 605: Why is a sharper gradient the basis for selection?
**Authors:** Thank you for asking this question. We feel that this question is similar to the previous comment "Line 150f: Why is the stronger gradient an argument for the use of the thermal tropopause? This is not clear to me. Did you consider the cold point tropopause (CPT) for the tropics?" Hence, we copy here the relevant text:
Thank you for the comment. In the revised paper, we have removed the sentence "Hence, due to the sharper gradients for the thermal tropopause, we will use this definition for the study". The paragraph now reads as follows:
"This analysis shows that the results are impacted by the choice of the tropopause definition. However, in this study, we are interested in the differences between IAGOS and ERA5, and are interested in the separation of dry and moist conditions. Although there are no significant differences in the results when using the thermal and dynamic tropopause definitions, there is a tendency for the dynamic tropopause definition to result in higher values of RHi in the lower stratosphere compared to the thermal tropopause definition. Therefore, we decided to use the thermal tropopause to ensure better separation of moist and dry conditions."

**Reviewer:** Figure A1: Saturation ratio or RHi?
**Authors**: Figure A1 has been adapted to show RHi with [%].

**Reviewer:** Figure S1: The important Information is very small in this figure, grey shading is not visible.
**Authors:** Thank you for pointing this out. We have modified Figure S1 and the grey shading should be more visible now. However, it should be noted that the accuracy of the IAGOS ICH sensor for temperature is ±0.5K, hence the shaded area is not very large. Subsequently, we have also improved Figure S2, S3 and S4.

**Reviewer:** Figure S6: What is the x-Axis? Is ΔRHi in units of RHi? Or is it a relative difference between RHi_iagos and RHi_era?
**Authors:** Thank you for asking this question. Indeed, there was a typo. The x-axis is the units of RHi, which we have also adapted this to be in [%].

**Works Cited**

Alduchov, O. A., & Eskridge, R. E. (1996). Improved Magnus Form Approximation of Saturation Vapor Pressure. *Journal of Applied Meteorology and Climatology*.

Gierens, K., Matthes, S., & Rohs, S. (2020). How Well Can Persistent Contrails Be Predicted? *Aerospace*.

Jolliffe, I. T., & Stephenson, D. D. (2012). *Forecast Verification - A Practitioner's Guide in Atmospheric Science.* John Wiley & Sons, Ltd.

Mehta, S. K., Ratnam, M., & Murthy, B. (2011). Multiple tropopauses in the tropics: A cold point approach. *Journal of Geophysical Research: Atmospheres*.

Murray, P. C., Finlayson, B., & McMahon, T. A. (2007). Updated world map of the Köppen-Geiger climate classification. *Hydrology and Earth System Sciences*. Retrieved from https://people.eng.unimelb.edu.au/mpeel/koppen.html

National Oceanic and Atmospheric Administration. (2023a). *Inter-Tropical Convergence Zone*. Retrieved from National Oceanic and Atmospheric Administration: https://www.noaa.gov/jetstream/tropical/convergence-zone

National Oceanic and Atmospheric Administration. (2023b). *JetStream Max: Addition Köppen-Geiger Climate Subdivisions*. Retrieved from National Oceanic and Atmospheric Administration: https://www.noaa.gov/jetstream/global/climate-zones/jetstream-max-addition-k-ppen-geiger-climate-subdivisions

Petzold, A., Khan, N. F., Li, Y., Spichtinger, P., Rohs, S., Crewell, S., . . . Krämer, M. (2025). Most long-lived contrails form within cirrus clouds with uncertain climate impact. *Nature Communications*.

Roosenbrand, E., Sun, J., & Hoekstra, J. (2023). Contrail minimization through altitude diversions: A feasibility study leveraging global data. *Transportation Research Interdisciplinary Perspectives*.

Schumann, U. (1996). On conditions for contrail formation from aircraft exhausts. *Meteorologische Zeitschrift* .

Schumann, U. (2012). A contrail cirrus prediction model . *Geoscientific Model Development*.

Schwartz, C. S. (2017). A Comparison of Methods Used to Populate Neighborhood-Based Contingency Tables for High-Resolution Forecast Verification. *Weather and Forecasting*.

Sonntag, D. (1994). Advancements in the field of hygrometr. *Meteorologische Zeitschrift*.

Teoh, R., Schumann, U., Gryspeerdt, E., Shapiro, M., Molloy, J., Koudis, G., . . . Stettler, M. E. (2022). Aviation contrail climate effects in the North Atlantic from 2016 to 2021. *Atmospheric Chemistry and Physics*.

Wang, Z., Bugliaro, L., Gierens, K., Hegglin, M. I., Rohs, S., Petzold, A., . . . Voigt, C. (2025). Machine learning for improvement of upper-tropospheric relative humidity in ERA5 weather model data. *Atmopsheric Chemistry and Physics*.

Wolf, K., Bellouin, N., Boucher, O., Rohs, S., & Li, Y. (2025). Correction of ERA5 temperature and relative humidity biases by bivariate quantile mapping for contrail formation analysis. *Atmopsheric Chemistry and Physics*.

---

## Author Comment (AC2)

**Reply to Reviewer #2**

We would like to thank the reviewer for their constructive comments, which have helped improve the paper significantly. We agree to shorten the paper to focus more on the new results, particularly those for the tropics. While reviewing the paper, we found an error in the analysis tool for the vertical distribution of the mean temperature was found and corrected (Fig. 7 in the preprint), i.e., the mean temperature was plotted with respect to the dynamic tropopause instead of the thermal tropopause. This only affected the interpretation of the results in Sect. 3.1, for which the corresponding findings has been corrected. However, it does not change the overall conclusion on the temperature comparison between ERA5 and IAGOS.

In the following, we reply to the individual comments from the reviewer. We use colour to organize this as follows:

- The questions and comments from the reviewers are marked in blue.
- The replies from the authors are written in black.
- Any changes to the manuscript are written in green.

Major Comments:
**Reviewer:** This study demonstrates a significant investment of time and effort, application of recognized methods, and extensive knowledge of IAGOS and ERA5, making the foundation of the study solid. However, the analysis itself is overly descriptive. Often minor differences between the old and new results are shown, regardless of their significance. Therefore, I recommend shortening the text and focusing more on relevant aspects.

**Authors:** Thank you for the feedback. The comparison between ERA5 and IAGOS indeed focuses a lot on confirming earlier studies, which is a way for us to evaluate our analysis approach. Nevertheless, we agree that the paper is perhaps too long. Following the suggestions, we have shortened the paper by focusing more on novel results, i.e. those from the tropics and removing comparisons to literature in the results section. Comparisons to literature are now mainly found in a newly added discussion section (Sect. 4). Furthermore, we have removed Sect. 2.5 on the North Atlantic weather pattern methodology and Sect. 3.6 on the analyses of the North Atlantic weather patterns based on other feedback received in the review process. These sections are now found in Sect. S7 in the supplement. We suggest it as future research in the discussion section. We have also removed Fig. 5 from Sect. 2.4, which showed the global mean annual cirrus occurrence, and Fig. 8 from Sect. 3.2, that showed a 2D histogram of the ERA5 RHi as a function of IAGOS RHi; we found that their key messages were mainly confirming previous results.

**Reviewer:** I don't think it makes much sense to calculate the ETS for many different situations when the poor agreement between the ISSRs found by ERA5 and IAGOS is due to the adjustment scheme under cloudy conditions. I would recommend repeating this part of the analysis by using a threshold of ERA5 RHi of 90 % or 95 % to determine ISSRs (at least under cloudy conditions).

**Authors:** We agree that repeating this analysis using different thresholds of ERA5 RHi to determine ISSRs is interesting. However, we find that only showing how the ISSR fraction changes with different thresholds might not give a good idea of whether different RHi thresholds improves the ability of ERA5 to predict ISSRs. This is because of how the ISSR fraction is calculated; we sum the number of counts where either IAGOS or ERA5 finds RHi > threshold and divide by the total number of measurements within the conditions considered (i.e. region, atmospheric layer, and in-cloud conditions). Thus, it does not consider if ERA5 improves at those specific points where it was either

not predicting an ISSR or incorrectly predicting an ISSR. This can be considered with the ETS. Hence, we have replaced the analysis in Sect. 3.5 with how the ETS changes with different RHi thresholds under cloudy and clear-sky conditions.

**Reviewer:** I also have to question the calculation of ETS in the context of discussing different weather patterns. Before discussing the differences in ETS, it would be helpful to examine the fraction of ISSRs for these weather patterns as observed by IAGOS, and determine if there are any differences among them.
**Authors:** We agree that examining the fraction of ISSRs for each weather pattern as observed by IAGOS and determine differences among them could be helpful. Now that we have removed Section 3.6 and included it as a discussion point, we have decided to include these figures in the supplement (see Sect. S7).

**Reviewer:** It is an interesting feature that in South Asia you find the minimum temperature 60 hPa below the thermal tropopause. The structure of the lapse rate and cold point tropopause in the tropics can be quite complicated but, normally, if there are multiple tropopauses the cold point tropopause is above the WMO thermal tropopause (also in Muhsin et al., 2018?). I suggest performing a literature review to learn more about this. See for example: https://agupubs.onlinelibrary.wiley.com/doi/epdf/10.1029/2011JD016637
**Authors:** Thank you for this comment and for the references. An error in the plotting tool for the vertical profile of mean temperature was noticed, where it was plotting the vertical profile with respect to the dynamic tropopause instead of the thermal tropopause (Fig. 7 in the preprint). It only affected the interpretation of the results in Sect. 3.1, for which the corresponding findings has been corrected. This has now been corrected to the thermal tropopause. The minima of temperatures in the tropic regions of Southern Trans-Pacific and Africa now appear more distinct. The minimum temperature in South Asia for JJA is no longer visible after fixing the error in the analysis tool. However, it does not change the overall conclusion that ERA5 shows a cold bias in temperature when compared to IAGOS. For example, we still find ERA5 to have a cold bias of 0.5 K on average in the extratropical upper troposphere.

**Reviewer:** Line 334 and 516: "Therefore, the moist bias appears to arise as a result of the saturation adjustment, which cannot take into account that the (wet) mode of the RHi PDF can be subsaturated in some regions." In my opinion this should not lead to a moist bias but to an alignment, because the (false) saturation adjustment would just not happen. Please explain.
**Authors:** We appreciate this feedback. We meant to say that it appears that ERA5 cannot consider that the wet mode can be subsaturated in cloudy conditions; under subsaturated conditions, ERA5 still showed a mean value of 100%, which is the value of the saturation adjustment. However, we do not know for certain why the subsaturation is not well estimated. We have rephrased the sentence to:
"This is because the mean RHi in IAGOS for this particular region and season is less than 100% in cloudy and indeterminate conditions, which aligns with the behaviour of the RHi PDF, whereas ERA5 shows a mean of 100%. Therefore, the moist bias appears to arise because ERA5 cannot take into account that the wet mode in cloudy conditions can be subsaturated in some regions. Whether the subsaturation is a result of the specific weather conditions in JJA for South Asia is uncertain and requires further evaluation."

Specific comments:

**Reviewer:** Line 123: In IAGOS RHL > 100 % is flagged invalid.
**Authors:** Thank you for pointing this out, this is a useful information.

**Reviewer:** Line 131: Why does sampling every minute result in 2.5% of all IAGOS measurements?
**Authors:** We can see that the structure of the paragraph was misleading. Indeed, it is not the sampling that results in 2.5% of all IAGOS measurements. We meant to present the percentage of measurements left after applying the criterion from Table 2 and the sampling together, but we realize the value presented is also wrong. This paragraph has been restructured, and the value has been updated. See the new paragraph below:
"Lastly, IAGOS records the measurements every four seconds. To avoid autocorrelation affecting our analysis, it is chosen to sample a measurement approximately every minute. This is done using a uniform random number generator ranging from 1 to the maximum number of measurement points from IAGOS, to avoid systematic bias. The sampling and application of the criterion from Table 2 results in using 3.8% of all IAGOS measurements between 01/07/2011 and 31/12/2022."

**Reviewer:** Line 182: You find a higher frequency of indeterminate and cloudy conditions compared to IAGOS because of the limitations of the BCP (see Petzold et al., 2017).
**Authors:** We agree, thank you for the confirmation.

**Reviewer**: Line 202: Yes, physically, relative humidity partially depends on temperature, but the ICH sensor from IAGOS measures RHL, so, this is the water variable which is independent from the temperature.
**Authors:** We have removed the sentence "This is important to consider, as relative humidity partially depends on temperature (Reutter et al., 2020)". However, we still consider temperature an important aspect to compare as it directly influences the selection of points considered for ice supersaturation, as mentioned in Section 3.2.

**Reviewer:** Line 205: Please specify how big the ERA 5 cold bias in the extratropics is.
**Authors:** We have specified how large the ERA5 cold bias is in the extra tropics.

**Reviewer:** Line 212: "extratropical seasonal cycle": in the Northern Hemisphere
**Authors:** This change has been incorporated.

**Reviewer:** Line 250: it does not provide good quality results in dry conditions in the lower stratosphere due to the loss of sensitivity as a result of the adiabatic compression effect (Konjari et al., 2025)
**Authors:** Thank you for this clarification. We have adapted the sentence to include "in the lower stratosphere".

**Reviewer:** Line 302: Please rephrase: "The tropopause layer is not completely dry (Petzold et al., 2020; Reutter et al., 2020)"
Petzold et al. state: "a tropopause layer characterized by mean RHice of 60% almost independent of the season..."
Reutter et al. state: "In the tropopause layer, still a significant amount of the data is exceeding values of RHi > 100 %, both in the in situ data as well as in the ERA data set."

**Authors:** We agree that this statement was not clear. At the same time, we have restructured this section to shorten the paper, but we still implement one of the suggestions. The suggestion has been included as follows:

"The UT has more moisture compared to the TROP and LS, allowing for higher probabilities of ISS occurrence. However, the TROP still shows high ISS occurrence, both in the tropics and extratropics. This is in line with Reutter et al. (2020), where it was found that there was a significant amount of in-situ and ERA-interim data that exceeded the RHi = 100% threshold near the tropopause in the North Atlantic flight corridor."

**Reviewer:** Line 314: "For clear-sky conditions, there is a smaller probability of low RHi, with increased probability of higher RHi, and a more equal probability across all observed RHi values." I don't understand this sentence.

**Authors:** We have rephrased this sentence to, while at the same time combining the analysis of the UT and TROP:

"For UT and TROP clear-sky conditions, the probability of RHi > 100% is higher, and there is a more uniform probability across the observed RHi range compared to the LS."

**Reviewer:** Line 341: typo: "ERA5 may predict less ISSRs compared to ERA5"

**Authors:** Thank you for pointing this out. With shortening this paper, this sentence has been removed.

**Reviewer:** Line 364: Delete the word "but" in: "On the other hand, Reutter et al. (2020) found a maximum of 40% using IAGOS when considering all seasons, but with the maximum also occurring 30 hPa below the tropopause and a reduction of the ISSR fraction to zero in the lower stratosphere.

**Authors:** We have restructured this paragraph, for which the 'but' has also been removed.

**Reviewer:** Line 367: "ISSR fraction is highly sensitive to…": Maybe better without the word "highly"

**Authors:** We agree, and we have removed the word 'highly' from this sentence.

**Reviewer:** Line 402: Could the reason for higher ETS in North America be, that there are more data assimilated for ERA5?

**Authors:** Thank you for asking this question. We have looked at the WMO WDQMS webtool to get an idea. Looking at this tool, for upper-air land observations from Near-real-time NWP monitoring of the Global Observing System networks, it does appear that more data is assimilated in North America compared to the North Atlantic and the tropical regions defined. However, it is hard to make that distinction between North America, Europe and North Asia. Hence, we do not think we can appropriately make this conclusion; we would require more accurate knowledge of how much data is assimilated for ERA5 in each region.

We contacted an expert focusing on data assimilation and differences between Europe and the United States, who mentioned that it is also a difficult question. The expert mentioned that there is more aircraft data over the United States compared to Europe and Asia. Furthermore, ECMWF also uses more METAR data in the US compared to Europe. Hence, the reason for a higher ETS in North America could be that more data is assimilated for ERA5, but we do not have enough information to guarantee this.

**Reviewer:** Line 407: Please rephrase: "the maximum difference between seasons is approximately 10%, at most. In North America, we find that the variation can be up to 20%."

**Authors:** Due to the restructuring and shortening of the paper, this sentence has been removed.

**Reviewer:** Line 410: "tend to find"

**Authors:** This change has been incorporated.